# Trajectory-centric framework TrajAtlas reveals multi-scale differentiation heterogeneity among cells, genes, and gene modules in osteogenesis

Litian Han[1]©, Yaoting Ji[1]©, Yiqian Yu[1], Yueqi Ni[1], Hao Zeng[1], Xiaoxin Zhang[1], Huan Liu [1,2,3]*, Yufeng Zhang [1,2,3]*

1 State Key Laboratory of Oral & Maxillofacial Reconstruction and Regeneration, Key Laboratory of Oral Biomedicine Ministry of Education, Hubei Key Laboratory of Stomatology, School & Hospital of Stomatology, Wuhan University, Wuhan, Hubei Province, China, 2 Frontier Science Center for Immunology and Metabolism, Wuhan University, Wuhan, Hubei Province, China, 3 TaiKang Center for Life and Medical Sciences, Wuhan University, Wuhan, Hubei Province, China

© These authors contributed equally to this work.
* liu.huan@whu.edu.cn (HL); zyf@whu.edu.cn (YZ)

**Data Availability Statement:** The Differentiation Atlas (raw counts, embedding, cell type annotations, biological and technical metadata) is publicly available and can be downloaded via

## Abstract

Osteoblasts, the key cells responsible for bone formation and the maintenance of skeletal integrity, originate from a diverse array of progenitor cells. However, the mechanisms underlying osteoblast differentiation from these multiple osteoprogenitors remain poorly understood. To address this knowledge gap, we developed a comprehensive framework to investigate osteoblast differentiation at multiple scales, encompassing cells, genes, and gene modules. We constructed a reference atlas focused on differentiation, which incorporates various osteoprogenitors and provides a seven-level cellular taxonomy. To reconstruct the differentiation process, we developed a model that identifies the transcription factors and pathways involved in differentiation from different osteoprogenitors. Acknowledging that covariates such as age and tissue type can influence differentiation, we created an algorithm to detect differentially expressed genes throughout the differentiation process. Additionally, we implemented methods to identify conserved pseudotemporal gene modules across multiple samples. Overall, our framework systematically addresses the heterogeneity observed during osteoblast differentiation from diverse sources, offering novel insights into the complexities of bone formation and serving as a valuable resource for understanding osteogenesis.

## Author summary

Osteoblasts are the cells responsible for bone formation, and they can originate from various types of progenitor cells. However, it is not well understood how these different progenitor cells become osteoblasts, or how factors such as age and tissue location influence this process. This knowledge gap is largely due to the lack of comprehensive tools that can

Figshare (https://figshare.com/articles/dataset/Differential_Atlas/25422688). The annotation system and OPC mapping of Differentiation Atlas can be interactively explored at https://zyflab.shinyapps.io/TrajAtlas_shiny/. The TrajAtlas software package is available at https://github.com/GilbertHan1011/TrajAtlas. The TrajAtlas documentation including API, tutorials, and examples is available at https://trajatlas.readthedocs.io/. Codes to reproduce our analysis are available at https://github.com/GilbertHan1011/TrajAtlasManuscript

**Funding:** This work was funded by grants from the National Key Research & Development Program of China (2021YFC2400405) to YZ; the National Natural Science Foundation of China (No. 82322014 and No. 82270948) and The Interdisciplinary Research Project of School of Stomatology Wuhan University (No. XNJC202306) and the Fundamental Research Funds for the Central Universities (2042022dx0003, 2042024kf1023) to HL. The funders had no role in study design, data collection and analysis, decision to publish, or preparation of the manuscript.

**Competing interests:** The authors have declared that no competing interests exist.

map and analyze the differentiation process. In this study, we introduce TrajAtlas, a novel comprehensive framework designed to fill this gap. TrajAtlas allows us to create a detailed map of osteoblast differentiation, track changes in gene activity, and examine the impact of age and tissue location on this process. Using TrajAtlas, we also identified patterns in gene expression that are consistent across different conditions, helping us uncover new cell states involved in bone healing. This framework not only enhances our understanding of osteoblast differentiation but also offers a versatile tool that could be applied to study how other types of cells differentiate in different contexts.

## Introduction

Osteoblasts, or bone-forming cells, play a critical role in the dynamic processes of bone formation [1,2], remodeling [2,3], and regeneration [4]. Due to their limited lifespan, osteoblasts require constant replenishment by osteoprogenitor cells (OPCs) [5]. This process, known as osteoblast differentiation [5], occurs in various tissues and different ages in response to various stimuli such as development and regeneration [5]. Across these tissues and age groups, diverse osteoprogenitors contributing to this process have been identified [5,6]. In this respect, recent studies have sequentially identified bone marrow stromal cells expressing markers *Lepr* [5–7], *Grem1* [8], and *Cdh2* [9] as osteoprogenitors in long bone. Besides, chondrocytes, expressing *Pthlh* [5,6,10], *Foxa2* [11], and *Fgfr3* [12], were also identified as cellular sources of osteoblast in long bone. The diversity of osteoprogenitor cells necessitates a systematic classification. However, the current approach based on immunophenotypic markers may be too crude to fully capture the big picture of osteoprogenitors [5,6].

The crude definition of osteoprogenitor cells has largely hindered our understanding of osteoblast differentiation. Multiple studies have proposed distinct pathways for osteogenesis in long bones, resulting in a fragmented view of the process [5,6,10,13,14]. While efforts to synthesize these findings into a unified framework have been undertaken [15], they failed to illuminate the intricacies of osteoblast differentiation. The current classification system, which dichotomizes osteogenesis into endochondral and intramembranous based on the presence of a cartilaginous template [16], overlooks the diverse cellular origins of osteoblasts. This oversimplification emphasizes the need for a novel model that incorporates the origin of osteoblasts, thereby providing a more accurate representation of osteoblast differentiation and addressing the existing knowledge gaps.

While cell origin may influence the differentiation process, there are many other covariates to consider, such as age [17,18], tissue [5,6], and injury [7], which may also contribute to heterogeneity. For example, studies have shown that the expression of certain genes, such as *Maf*, declines with age [17]. Conversely, under injury conditions in long bones, the Wnt signaling pathway is upregulated to promote bone regeneration [7]. The influence can occur in different aspects, including gene expression [19], transcription factors [20], and pathways [20], and these changes may vary throughout the differentiation process. For example, the early stages may differ significantly from the later stages [21]. These factors render differentiation a complex puzzle and make it difficult to fully grasp the concept.

In recent years, single-cell technologies have revealed cellular states with unprecedented detail, providing insights that facilitate the reconstruction of differentiation processes. Numerous algorithms have emerged, each aiming to reconstruct differentiation from different perspectives [22,23]. However, these methods are often isolated and lack a comprehensive view of differentiation. Each method raises distinct questions—such as the sequence and fate of cells

during differentiation (Monocles [22], Slingshot [23]), the changes in gene expression patterns over time (tradeSeq [19]), and the influence of covariates on this process (Lamian [21], Condiments [19]). However, to fully understand the dynamics of differentiation, it is essential to examine cells, genes, and gene modules across the diverse spectrum of differentiation trajectories. This limitation underscores the need for more sophisticated analytical frameworks that can integrate and interpret the vast and intricate data generated by single-cell technologies, ultimately providing a more nuanced and holistic understanding of cellular differentiation processes.

To address the complexity and heterogeneity inherent in osteoblast differentiation, our study introduces **TrajAtlas**, a trajectory-centric framework specifically designed to navigate and elucidate these challenges. TrajAtlas encompasses a reference atlas for osteoblast differentiation coupled with a sophisticated seven-layer hierarchical annotation system that allows for the exploration of osteoprogenitor cell heterogeneity. In-depth, a model was introduced to reveal genetic regulation and differentiation pathways from diverse osteoprogenitor cells to mature osteoblasts. It features advanced tools for dissecting cell and gene dynamics along trajectories, facilitating the identification of age and tissue-specific variations and uncovering differentiation-related gene modules to infer their activity within these paths. By leveraging TrajAtlas, with a particular focus on the influences of age and injury on bone formation, our framework unveils novel insights into osteoblast differentiation. Importantly, while our framework is exemplified through the study of osteogenesis, it is designed to be broadly applicable. When supplemented with single-cell transcriptome datasets from other tissues or lineages, TrajAtlas can be adapted to reveal similar insights into the differentiation processes of various cell types, making it a versatile tool for understanding cellular differentiation across different biological contexts. We have made the associated datasets available for interactive exploration at https://zyflab.shinyapps.io/TrajAtlas_shiny, and the corresponding computational framework as open-source software, available at https://trajatlas.readthedocs.io.

## Results

### Overview of TrajAtlas

Cell differentiation is a complex problem involving changes in cells, genes, and gene modules. To address this complexity in a more comprehensive way, TrajAtlas comprises four distinct modules, enabling the multi-scale exploration of differentiation heterogeneity. (a) **Heterogeneity of Osteoprogenitor Cells.** The first module, the Differentiation Atlas, integrates trajectories across diverse tissues and ages to create a comprehensive reference atlas of osteoblast differentiation, covering 110 samples and 272,369 cells (S1 Table). A seven-level hierarchical annotation system reconciles conflicting osteogenesis-related cell types across different studies. By mapping experimentally validated osteoprogenitor cells onto this atlas, we can more thoroughly investigate their heterogeneity (Fig 1A). (b) **Genetic Regulation and Pathways Driving Osteoprogenitor Differentiation**: the second module, the Differentiation Model, reconstructs the osteoblast differentiation process. It employs four trajectory groups to represent the differentiation pathways from various osteoprogenitor cell populations, offering insights into the critical signaling pathways and gene regulatory networks that govern this process (Fig 1B). (c) **Age and Tissue-Specific Variations in Differentiation**: the third module, TrajDiff, facilitates the detection of covariate-associated differential cell abundance and gene expression along the differentiation process across multiple trajectories. This tool allows for the assessment of factors such as age and tissue-specific location on gene expression patterns during osteoblast differentiation (Fig 1C). (d) **Identification of Gene Modules Related to Differentiation**: the fourth module, TRAVMap, introduces novel methods for identifying robust

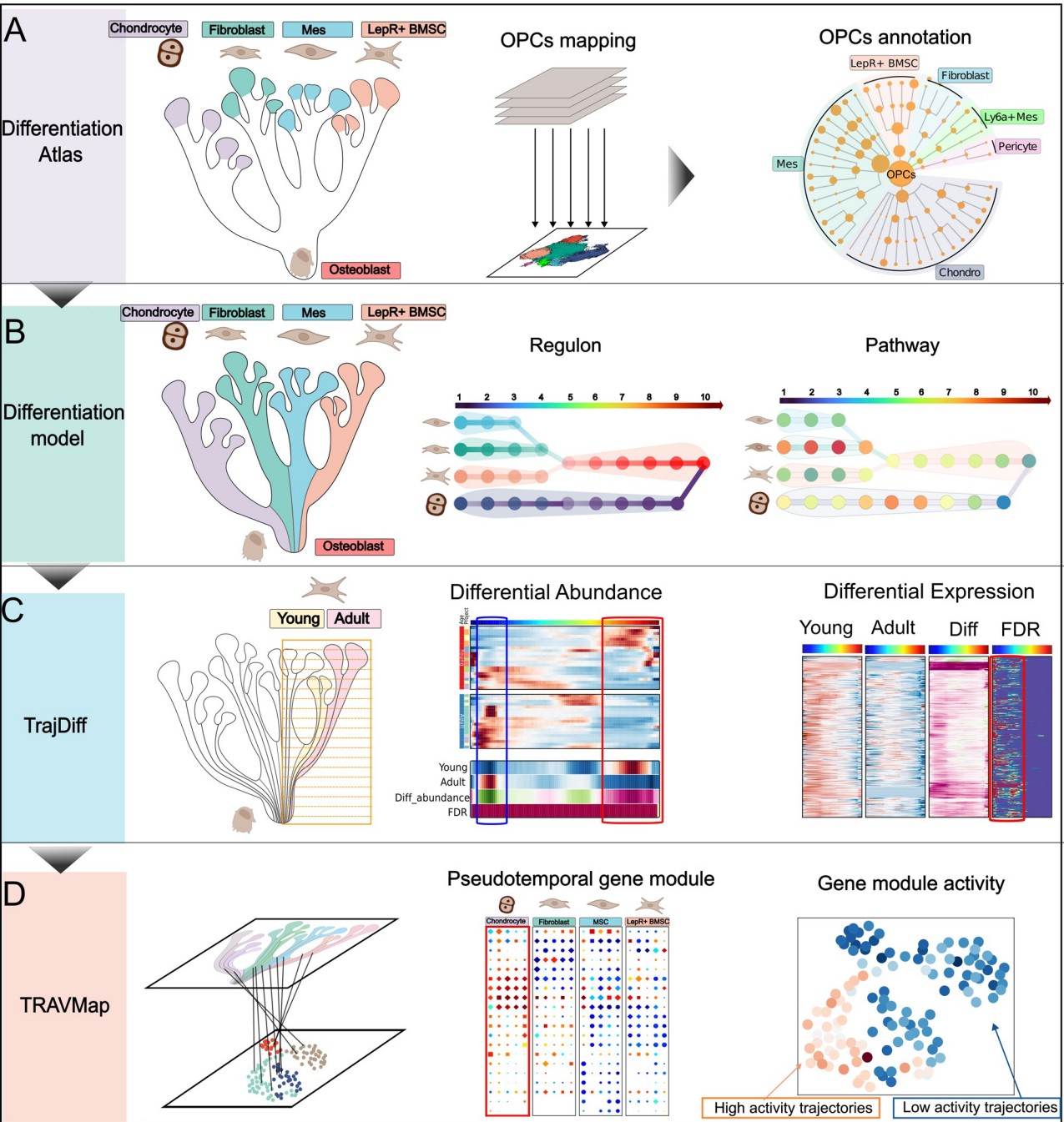

**Fig 1. Overview of TrajAtlas: A framework designed to unravel osteogenesis heterogeneity in a multi-scale manner. (A) Differentiation Atlas** integrates trajectories spanning various tissues and continuous age groups to construct a differentiation atlas aimed at identifying various osteogenic precursor cells (OPCs). **(B) Differentiation Mode**l reconstructs the osteoblast differentiation process, unveiling key genes and transcription factors associated with OsteoProgenitor Cell-Specific Trajectory (OPCST). (**C**) **TrajDiff** detects covariates-related differential cell abundance and gene expression along the differentiation process across multiple trajectories. (**D**) **TRAVMap** module identifies trajectory-related gene modules and infers gene module activity across large-scale trajectories. This figure was created with BioRender.com.

gene modules across multiple datasets. It enables the investigation of the roles these gene modules play within large-scale differentiation trajectories and their relationship to differentiation

processes (Fig 1D). Through these approaches, we enable the unraveling of differentiation heterogeneity in cells, genes, and gene modules.

## The osteogenic differentiation atlas reveals the heterogeneity of osteoprogenitor cells

To construct a comprehensive map of osteoblast differentiation, we combined 26 public datasets into the **Differentiation Atlas**, encompassing a total of 272,369 cells (Fig 2A). These datasets originated from three primary osteogenic tissues (head, limb bud, and long bone [5,6]) across various age groups, ranging from embryo to old age (Methods). With a focus on the differentiation process, we filtered out cells irrelevant to osteogenesis (Methods). Next, we employed **scANVI** [24] to integrate the datasets, preserving biological variations while removing batch effects within the atlas (Figs 2A and S1A–S1D and S2 and Methods). Subsequently, we implemented a multi-level clustering method to visualize the hierarchical organization of cell populations (Figs 2C and S3 and Methods). We manually annotated the first three levels of clusters based on previous studies [5,6,25] and then utilized marker genes to distinguish finer cell states (Figs 2C and S4 and Note 1 in S1 Text). Furthermore, we harmonized findings from previous studies within the atlas and provided detailed descriptions of most clusters in the supporting information and online websites (S2 Table and Methods). With the extensive curated atlas and detailed annotations, we were able to explore the heterogeneity of osteoprogenitor cells.

We found that osteoprogenitor cells exhibited diverse distributions [6] and expressed distinct characteristic markers (Fig 2C), as reported in previous studies [5,6]. We curated osteoprogenitor marker genes, tissue locations, ages, and validation methods from 28 studies on osteoprogenitors, the majority of which validate lineage tracing in vivo, providing the highest level of biological evidence (23 out of 28) (S3 Table). To assess the osteogenic potential of cells within our atlas, we incorporated experimentally validated osteoprogenitors based on previously described marker genes and tissue locations (Fig 2A). Notably, we observed that osteoprogenitor cells exhibit great diversity, and rather than being distinct isolated states, they may be interconnected (Fig 2A and 2B). Utilizing our level-2 annotation system, we categorized these osteoprogenitor cells into six major cell types: chondrocytes, mesenchymal cells (Mes), *Ly6a* + Mes, LepR+ bone marrow stromal cells (LepR+ BMSCs), fibroblasts, and pericytes (Fig 2C).

Chondrocytes, a major cellular source of osteoblasts in the growth plate [26] were primarily located in the limb bud and long bone datasets within our atlas (Fig 2C and 2D). Chondrocytes can be further refined into four distinct clusters, all of which are reported to be osteoprogenitor cells with different spatial-temporal distributions. Chondrocyte progenitor cells (CPCs) typically express *Grem1* [25] and *Pthlh* [27], which may located in the resting zone of the growth plate [5,27] (Figs 2C and S5B). *Hmmr*+ CPCs [10] represent proliferating osteoprogenitors in the articular cartilage and the growth plate (Figs 2C and S5A). Hypertrophic chondrocytes (HCs) [5,28] marked by *Col10a1* can directly differentiate into osteoblasts (Fig 2C). Mature chondrocytes expressing *Foxa2* [11] may contain long-term osteoprogenitors involved in growth plate regeneration (Figs 2C and S5C). Interestingly, the limb bud predominantly contained *Hmmr*+ CPCs, while the head exhibited a higher proportion of CPCs (Fig 2C). This distribution highlighted the potential influence of tissue origin and developmental stage on these subpopulations.

Mesenchymal cells (Mes), characterized by the expression of *Prrx1* [29], are the main cellular source in embryo head and limb bud. They compose a large population in head and limb bud datasets in our atlas. As age increases, the cellular population of Mes declines and their cellular states undergo great variation [29] (Figs 2C and 2D and S6A and S6B). We annotated these states according to their prominent age as follows: Early Mes, predominant during the organogenesis

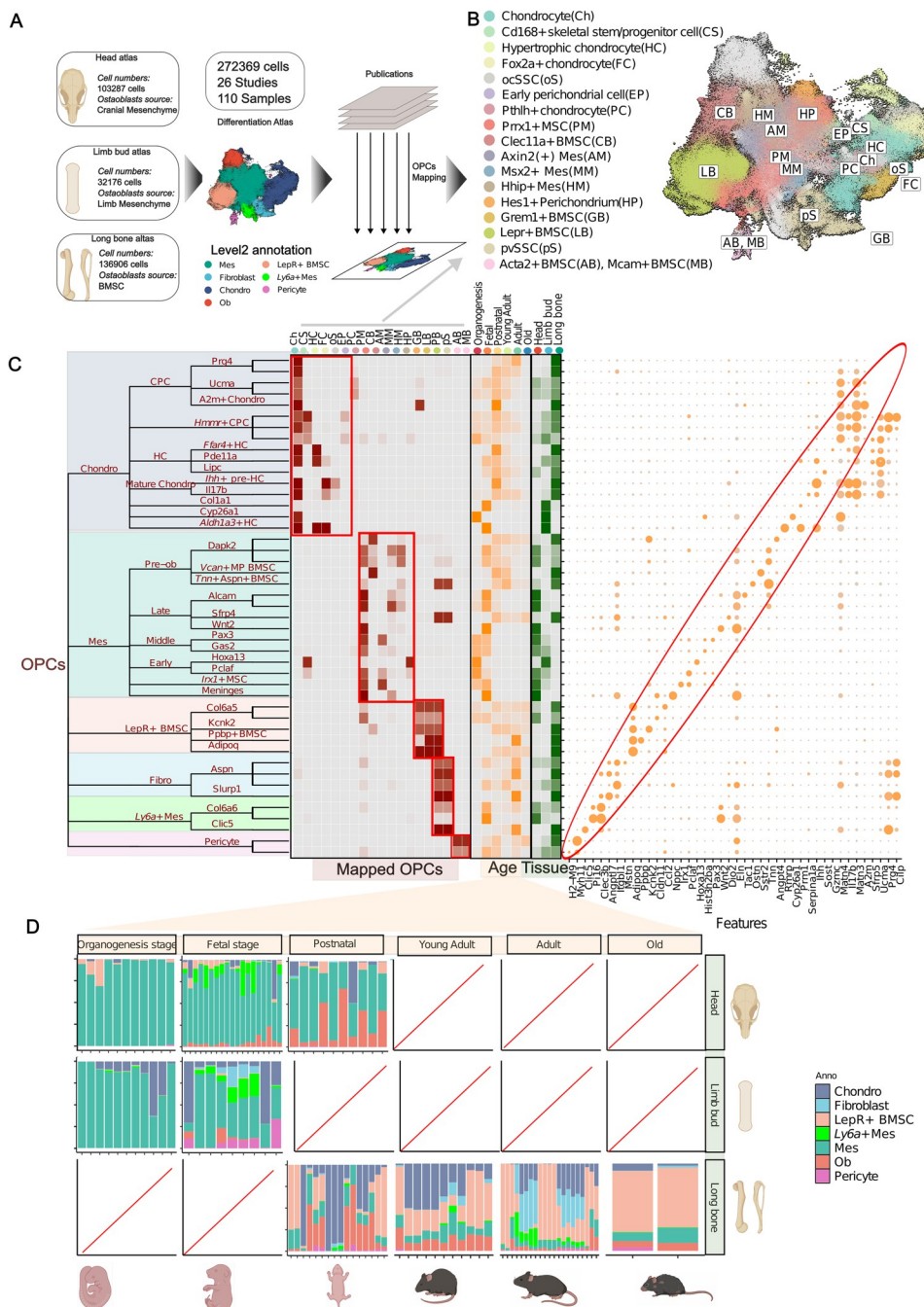

**Fig 2. Differentiation atlas reveals the heterogeneity of Osteoprogenitor cells. (A)** A schematic for differentiation atlas. **(B)** UMAP visualization of experimentally validated osteoprogenitors mapped to **Differentiation Atlas**. **(C)** A hierarchical tree of clusters of Differentiation **Atlas**. The first five levels with up to 49 clusters are presented, emphasizing the diverse nature of osteoprogenitors across various tissues and age groups. The left heatmap (red) depicts the overlapping of experimentally validated osteoprogenitors with clusters in the lowest tree level in the **Differetiation Atlas**. The middle heatmap (orange) depicts the relative percentage contribution of each cluster at the lowest tree level to the age group. The right heatmap (dark green) illustrates the relative percentage contribution of each cluster to the tissue origin group. In the right panel, dotplot displays marker genes at level 5 (Methods). **(D)** Barplots illustrates the proportions of cell types annotated with level-2 annotation across 65 samples with different tissue and age groups. Panel **(A)** and **(D)** were created with BioRender.com.

stage (E8.5-E14), manifested the early perichondrial marker *Hes1* [30] (Fig 2C); Middle Mes, the primary cell population in both organogenesis and fetal stage (E14.5-E18.5), specifically exhibited the suture mesenchyme marker *Axin2* [31] (Figs 2C and S5D); Late Mes, the major cell population in the postnatal stage (P0-P30), highly expressed *Msx2* [32] and may represent osteoprogenitors in specific craniofacial regions (Fig 2C). Besides, we found a cell state that highly expressed *Ly6a* (stem cell antigen-6), known markers of progenitor-like cells (S5E Fig). This population was notably present during the fetal stage in the head and limb regions (Fig 2D), as described in previous studies [29,33]. Consequently, we termed this cluster *Ly6a*+ Mes [33].

LepR+ BMSC cells, well-known for their roles in maintaining the bone marrow vasculature and regulating hematopoiesis [5–7], are multipotent cells residing within long bones [34]. Our atlas revealed LepR+ BMSCs in long bones only after the postnatal stage, with their population expanding as the organism ages (Fig 2D). Interestingly, *Grem1*+ BMSC [8] and *Cdh2*+ BMSC [9] identified in previous studies demonstrated significant overlap with LepR+ BMSC in our atlas (Fig 2B).

Fibroblasts, characterized by high expression of *S100a4* [25], were primarily observed in the adult stage (3M-12M) of long bones (Fig 2D). We found that they expressed periosteum maker, such as *Postn* [35], indicating their potential as a source of osteoblasts (Figs 2C and S7L and S7M). Pericytes, characterized by high expression of *Acta2* [25] (Fig 2C), constituted a minor population within our atlas (Fig 2D).

Our observations suggest that age is likely the most significant factor influencing the cellular states of osteoprogenitors (S6A and S6B Fig). To further investigate this, we performed a gene-level analysis (Methods). Interestingly, we found that most genes displayed consistent age-related effects across all osteoprogenitor types (S6C and S6D Fig). For instance, genes associated with bone formation, such as *Bmp2*, *Chrdl1*, and *Itgb3*, were upregulated with increasing age across all osteoprogenitors, while cell-cycle-related genes like *Cdk8* and *Parp1* were downregulated (S6C Fig). GSEA enrichment results indicated that pathways related to oxidative stress, hypoxia response, and lipid synthesis were upregulated across all osteoprogenitors with increasing age. Conversely, cell cycle activity, mRNA splicing, the Wnt pathway, and metabolism of glycine, serine, and threonine were downregulated (S6E Fig). These findings align with a previous study [36], indicating a conserved effect of age on osteoprogenitors, which is potentially linked to bone disorders like osteoporosis.

## Differentiation Model provides a comprehensive understanding of osteogenic differentiation

To understand how various osteoprogenitor cells differentiate into osteoblasts differentially, we established a **Differentiation Model** that focuses on the genetic regulation and pathway dynamics during osteogenesis. In our Differentiation Atlas, a continuum of cellular states between osteoprogenitors and osteoblasts in both UMAP and force-directed graphs was observed, indicating that our atlas reconstructs cellular transitions during osteoblast differentiation (Figs 2A and 3A–3C). Besides, we observed that different samples shared similar transition processes from specific osteoprogenitors, supporting the grouping of these trajectories together for further analysis (S8A–S8C Fig). To distinguish these grouped trajectories from individual sample trajectories, we termed them OsteoProgenitor Cell-Specific Trajectories (OPCSTs). We first investigated which osteoprogenitor clusters could directly transition into osteoblasts. To achieve this, we employed coarse-grained connectivity structures to capture cell transitions [37] between osteoprogenitors and osteoblasts, with higher connectivity indicating a greater probability of transition. (Figs 3A and S8D). This approach identified four

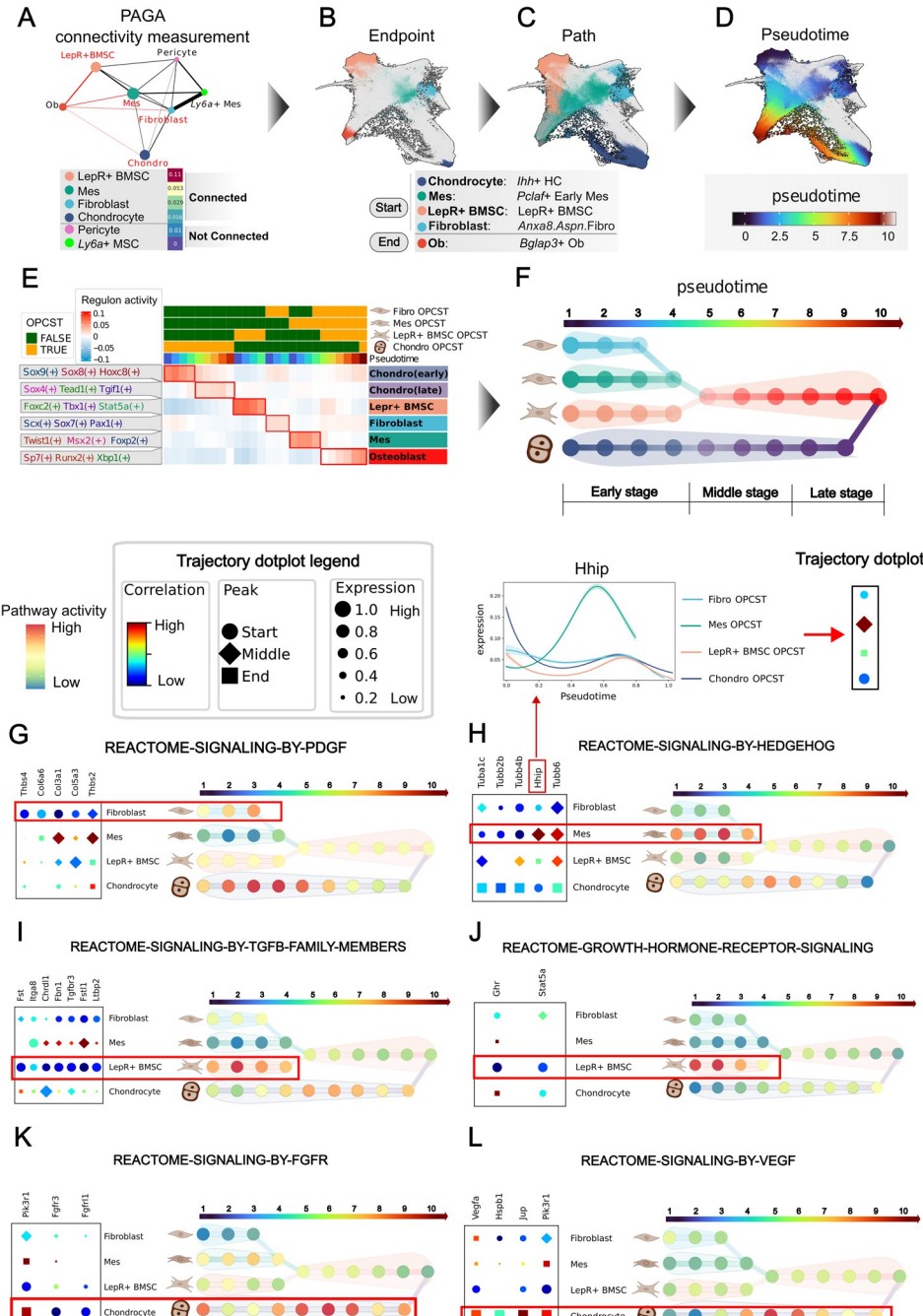

**Fig 3. Differentiation Model provides a comprehensive understanding of osteogenic differentiation. (A)** PAGA captures the cell transition of four osteoprogenitor clusters to osteoblasts through coarse-grained connectivity structures. (**B**) Force-directed graph of differentiation atlas colored by the endpoint of the four osteoprogenitor clusters. (**C**) Force-directed graph of differentiation atlas colored by differentiation path from the four osteoprogenitor cell-specific trajectories (OPCST). (**D**) Force-directed graph of differentiation atlas colored by common pseudotime of the four OPCST. (**E**) Heatmap showing the activity of six regulon clusters (row) across different pseudotime bins of four OPCST (column). (**F**) **Differentiation Model** reconstructs the differentiation process of osteoblast. Each node represents a pseudotime bin of OPCSTs, and colors represent regulon clusters in (**E**). pseudotime bins with similar scANVI representation are merged. (**G-I**), Activity of key pathway across four OPCSTs. The right panel depicts a trajectory dotplot of genes in the pathway, where the colors represent the correlation coefficient between gene expression and pseudotime; the size of the dots represents gene expression along the pseudotime; the shape indicates the pseudotime where maximum expression occurs. The left panel illustrates the pathway activity across four OPCSTs. The color represents pathway activity. Panel (**F**) was created with BioRender.com.

distinct grouped trajectories: Chondrocyte OPCST, LepR+ BMSC OPCST, Fibroblast OPCST, and Mes OPCST (Figs 3A and S8D).

Next, to identify the endpoints of the developmental trajectories, we explore the earliest states of osteoprogenitors (Figs 3B and S9A–S9C and Note 3 in S1 Text). To determine a universal indicator of differentiation from various osteoprogenitors, we benchmarked multiple machine learning strategies and ultimately employed an LGBMR Regressor to build common pseudotime (Figs 3D and S10A–S10H and Note 4 in S1 Text). To build the final model, we divided the differentiation path based on osteoprogenitors and pseudotime into bins, then merged bins representing similar cell states (Figs 3F and S11B–S11H and Methods). Furthermore, we conducted gene regulatory network inference to predict potential transcription factors and infer the activity of key pathways involved in osteogenesis (Figs 3E–3L and S12A–S12L and Methods). Additionally, we created a trajectory dotplot to visualize the pseudotemporal expression patterns across multiple trajectories (Figs 3G–3L and S13A–S13I and Note 5 in S1 Text). The inferred trajectory path of the **Differentiation Model** shows significant overlap with the reported lineage tracing cells [7,28], and most of the inferred transcription factors (82/136) have been demonstrated to be associated with bone formation (S12M Fig and S5 Table). Besides, a large portion of differentiated genes (938/1852) have been annotated in bone databases [38–40], such as Phylobone (S14Q Fig and Methods and S6 Table), further validating the model's ability to capture biologically relevant information about bone development.

Previous research has described the cellular transition between chondrocytes and osteoblasts [28]. This transition can be well illustrated in our model, represented as Chondrocyte OPCST (Fig 3A). In our model, the Chondrocyte OPCST primarily progressed through three stages: prehypertrophic chondrocytes, hypertrophic chondrocytes, and osteocytes [25,28] (Fig 3E). In this differentiation process, transcription factors and signaling pathways undergo significant changes. In terms of transcription factors, early differentiation is regulated by *Sox9*(+) and *Sox8*(+), which maintain the chondrocyte fate, while late differentiation is regulated by *Tgif1*(+) and *Sox4*(+), both of which are associated with parathyroid hormone [41,42] (Figs 3D and S12G and S12H). As for signaling pathways, PDGF signaling (*Col9a1*, *Col6a3*, *Thbs3*) is highly activated in the early differentiation process, while FGFR signaling (*Fgfr3*, *Fgfrl1*, *Pik3r1*) and VEGF signaling (*Vegfa*, *Jup*, *Hspb1*) are involved in the late differentiation process (Fig 3K and 3L). These results show a significant reprogramming in chondrocytes during the transition toward osteoblast differentiation.

In long bones, LepR+ BMSC cells are typically dormant but become osteogenesis-activated in response to injury [7]. Our model divides this process into three stages: LepR+ BMSC cells, pre-osteoblast cells, and osteoblasts (Figs 3A and S8D). Like Chondrocyte OPCST, we observed that transcription factors and pathways exhibited cascading changes. *Foxc2*(+) [43] and *Stat5a*(+) [44] regulate the early stage of differentiation (Figs 3D and S12E and S12F), while *Sp7* and *Runx2* regulate the late stage (Figs 3D and S12K and S12M). TGF-β signaling and growth hormone receptor signaling pathway were highly activated at early differentiation, while extranuclear estrogen signaling at late (Figs 3I and 3J and S14L). Apart from LepR+ BMSC, fibroblasts have attracted more and more attention in long-bone regeneration [35]. Our model delineated the Fibroblast OPCST, which illustrates this differentiation process in fibroblasts (Fig 3A–3D and S8D). Different from LepR+ BMSC, *Pax1*(+) [45] and *Scx*(+) [46] regulate the early differentiation, and the PDGF signaling pathway (*Thbs4*, *Thbs2*, *Col6a6*) was highly active during this stage (Figs 3D and 3G and S12A and S12B).

Mesenchymal cells are the main cellular source of both limb buds and heads in the embryo. Our model captured this process and depicted it as the Mes OPCST. At early differentiation, *Twist1*(+) [29] and Msx2(+) [32] were transcription factors that regulate this process, and the Hedgehog signaling pathway (*Hhip*, *Tubb6*, *Tubb4b*) exhibited high activity at this stage

(Figs 3D and 3H and S12C and S15D). Interestingly, we found that mesenchymal cells differentiated to pre-osteoblasts sharing similar cellular states with pre-osteoblasts in LepR+ OPCST (S11E Fig). However, pre-osteoblasts in Mes OPCSTs demonstrated high activity in the Wnt signaling pathway (*Wnt6*, *Wnt10a*, *Wnt2*), whereas pre-osteoblasts in LepR+ BMSC OPCSTs exhibited high activity in the Complement Cascade (*C3*, *C4b*, *Cfh*) (S14N–S14P Fig). This difference suggested that different osteoprogenitors could retain part of their own identity for responding to the microenvironment even after transitioning into similar cell states.

Overall, our **Differentiation Model** reconstructs the osteoblast differentiation process from different osteoprogenitors and provides a systematic view of the diversity in gene regulatory networks and pathway activities during osteoblast differentiation.

## TrajDiff detects covariate-related gene differences across multiple trajectories

It is now understood that throughout the process of differentiation, variations in both cell abundance and gene expression [21,47] are highly influenced by factors such as age [17] and tissue origin [48]. In multi-stage differentiation processes like osteoblast differentiation [3] (Fig 3F), it is pivotal to pinpoint the stage at which differential genes exert their influence. However, current methods [21,47] fail to infer the differential abundance and expression within specific differentiation stages, thus calling for the development of new tools. In the present study, we introduced **TrajDiff**, a novel tool specifically designed for performing differential pseudotime analysis across multiple samples, aimed at uncovering changes in cell abundance and expression throughout differentiation stages (Fig 4A). This algorithm enables us to detect age and tissue-specific variations in differentiation. Our benchmarking results demonstrated that **TrajDiff** could outperform existing algorithms, such as **Lamian** [22] and **Condiments** [47], in various aspects, including the detection of differential abundance (S15A–S15I Fig and Note 6 in S1 Text) and trend differences of gene expression (S16 and S17 Figs and Note 6 in S1 Text).

Our previous analysis on osteoprogenitors indicated that the states of LepR+ BMSCs and mesenchymal cells could change with age (S6A–S6E Fig). Additionally, prior studies have shown significant remodeling of gene expression [49] and biological function [34, 49] in LepR + BMSCs during adolescence. This evidence suggested that the differentiation process of LepR + BMSC may vary across different ages. To explore this age-related heterogeneity, we applied **TrajDiff** to LepR+ BMSC OPCST. Samples younger than 3 months were defined as the "Young" group, while others were assigned to the "Adult" group, based on previous studies [34,49] (Fig 4B). Differential abundance analysis revealed higher cell density at the late stage (preosteoblast and osteoblast) in the Adult group compared to the Young group, where cell density was higher at the early stage (LepR+ BMSC) (Fig 4C and 4D). This observation was unlikely to be due to technical sampling bias, as it has been consistently observed across multiple projects (Fig 4D). These results suggested that LepR+ BMSC cell abundance was low during postnatal and adolescent stages, which might explain why LepR+ BMSCs are not the main source of osteoblasts in the long bones of young mice during adolescence [34].

Using **TrajDiff**, we next performed differential expression analysis to identify genes whose expression varied across differentiation stages. Our study identified 74.2% of the differentially expressed genes reported in the previous study [49] (S18J Fig). While the previous study provided a binary classification of upregulated or downregulated genes [49], our results offer a more comprehensive understanding by elucidating the precise developmental stages at which these genes exhibit differential expression patterns. We categorized these differentially expressed genes into eight groups based on their expression patterns (S18B–S18J Fig). We

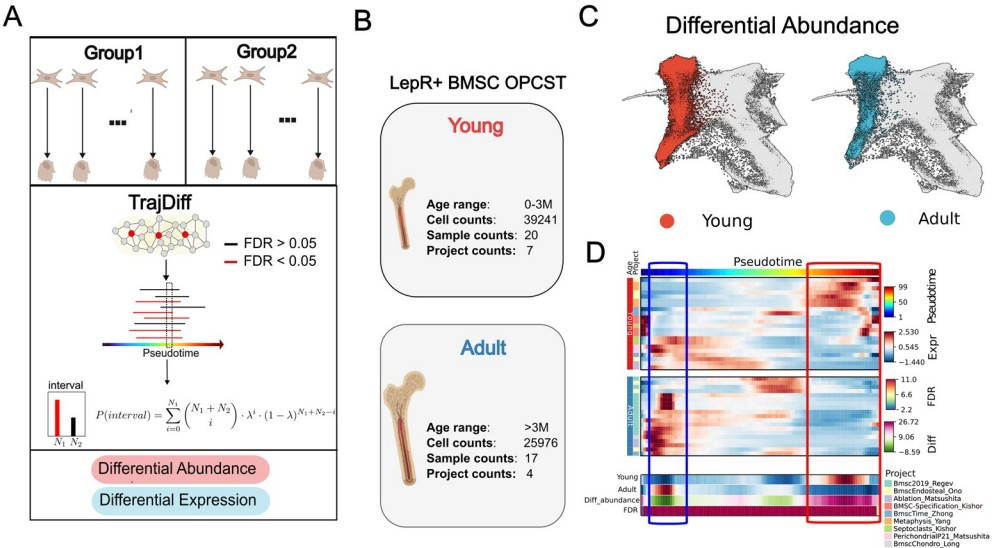

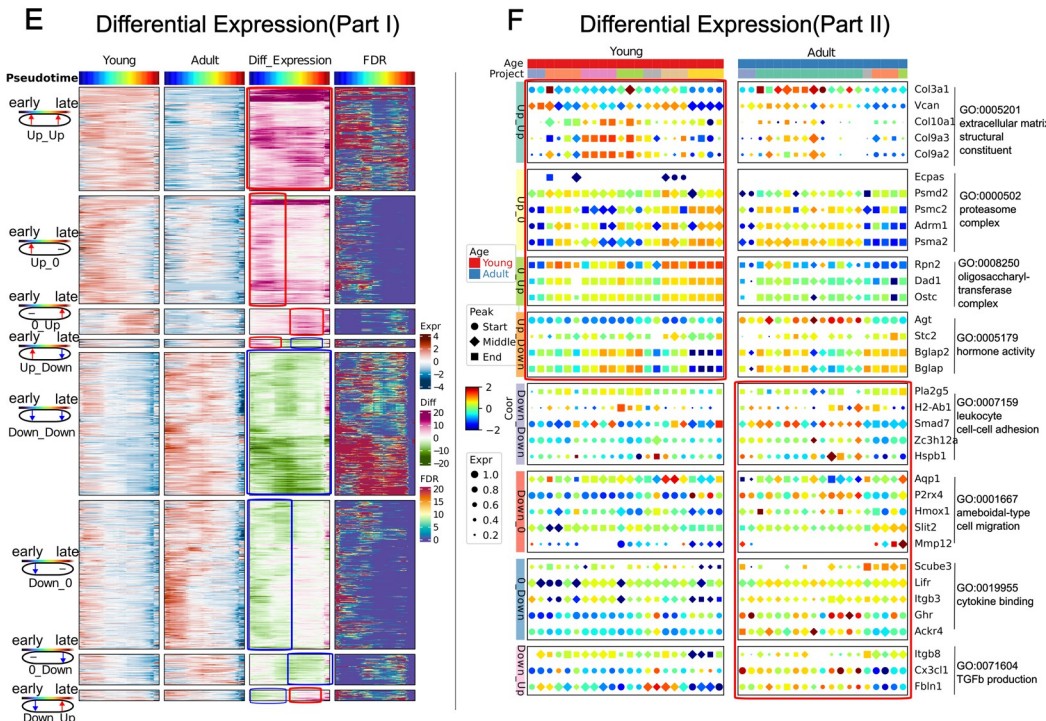

**Fig 4. TrajDiff detects covariate-related gene differences across multiple trajectories. (A)** A overview for **TrajDiff**. **(B)** Study design for **TrajDiff** analysis. **(C)** Force-directed graph visualization illustrates the difference in cell abundance between the Young and Adult groups. **(D),** Heatmap illustrates the difference in cell abundance along pseudotime (column) in 37 samples (row) between two groups. The four rows of bottom annotation represent: the mean cell abundance of the two groups (row 1, row 2), differential abundance (row 3), and false discovery rate (FDR) (row 4). **(E)** Heatmaps are presented in four vertical panels to illustrate gene expression of the Young group (first panel) and Adult group (second panel), expression differences between two groups (third panel), and FDR (fourth panel). In each panel, rows represent genes, while columns represent pseudotime. Genes are categorized into 8 clusters based on their expression patterns. **(F),** Trajectory dotplot illustrates expression of GO-enriched genes from eight gene clusters in **(D)** across 37 trajectories. Panel **(A)** and **(B)** was created with BioRender.com.

noticed that some genes are differentially expressed during the whole differentiation process. For example, genes related to leukocyte cell–cell adhesion, such as *Smad7*, *Hspb1*, and *Pla2g5*, showed higher expression in the Adult group than the Young group during differentiation ("Down_Down") (Figs 4D and 4E and S18A and S18G). While some genes are only differentially expressed at the specific differentiation process. Genes involved in ameboidal–type cell migration (*P2rx4*, *Mmp12*, *Slit2*), for instance, were downregulated only at the early stage ("*Down_0*") (Figs 4D and 4E and S18A and S18F). Interestingly, we noticed that certain genes were highly expressed in one group during early differentiation, but then became highly expressed in the other group during late differentiation. Genes related to TGF-β production (*Itgb8*, *Cx3cl1*, *Fbln1*) and hormone activity (*Agt*, *Bglap*, *Bglap2*) are both of this type, which suggests the roles of signaling pathways vary depending on age (Figs 4D and 4E and S18A, S18E, S18H and S18I). These findings corroborate previous research that reported enrichment of pathways associated with hematopoiesis, inflammatory responses [49], and antigen processing in older mice, whereas our results provide higher resolution insights across developmental stages.

Next, we investigated genes with transient differential expression. These genes, representing a smaller proportion (357 genes) compared to persistently differentially expressed genes (876 genes), exhibited dynamic changes throughout differentiation (S17A and S17C Fig). In the Young group, transiently upregulated genes at the early stage were associated with neuronal activity (*Clstn2*, *Flrt3*, *Shisa9*) (S17A and S17B Fig). Conversely, genes transiently downregulated at the early stage were associated with morphogenesis (*Sox9*, *Lhx1*, *Spg11*) (S17A and S17B Fig). Additionally, *Hes1*, involved in cell differentiation, was downregulated at the middle stage, while *Stk17b*, associated with fibroblast apoptosis, was downregulated at the late stage (S17A and S17B Fig).

In summary, **TrajDiff** offers a distinct advantage over **Lamian** and **Condiments** by identifying differential genes at precise developmental stages. Consequently, it furnishes a comprehensive landscape of age-related gene expression dynamics during the differentiation of LepR + BMSCs.

## TRAVMap reveals pseudotemporal gene module heterogeneity across all trajectories

During differentiation, groups of genes that participate collectively to execute specific functions are referred to as gene modules [50]. However, current gene module detection methods can't integrate multiple datasets to yield more reliable results. To address this limitation, we developed TRAVMap, a method for identifying differentiation-related gene modules across multiple trajectories (Fig 5A). TRAVMap utilizes pseudotime as an axis, and by projecting gene expression onto this axis, it identifies recurring axes of variation [51] within these trajectories, termed Trajectory-related Replicable Axes of Variation (TRAVs) (Fig 5A). We found that genes that drive TRAVs show coordinated expression patterns, representing gene modules that function at specific stages of the differentiation process (S20D, S20F and S20H Fig). Additionally, these gene modules are conserved across multiple samples, demonstrating the robustness and effectiveness of our algorithm (S20H Fig). TRAVMap distinguishes itself from existing gene module detection algorithms in two key ways: (1) it integrates multiple samples to derive more robust gene modules, and (2) it focuses specifically on pseudotemporal dynamics, offering insights into gene function across the temporal progression of differentiation. In this way, **TRAVMap** identified 226 TRAVs across 121 trajectories (Fig 5B). Furthermore, by leveraging TRAVs and gene expression data, TRAVMap learns representations of trajectories to capture sample-to-sample variability, allowing us to identify the sources of variability. Unlike current deep learning approaches [52], our approach incorporates gene and gene

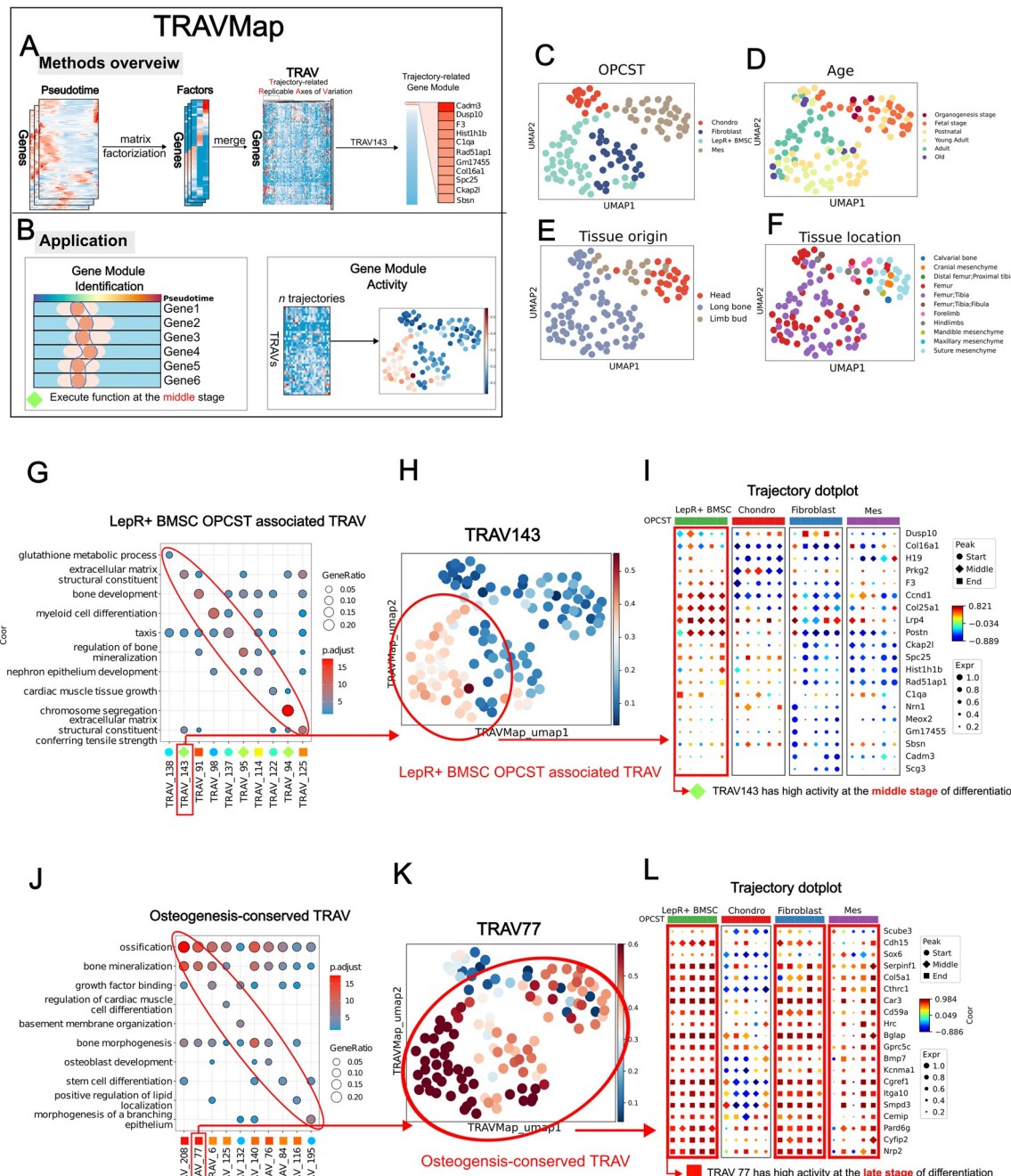

**Fig 5. TRAVMap reveals gene module heterogeneity across all trajectories. (A,B)** An overview for **TRAVMap**. **(B-E)** Trajectory embeddings visualized with UMAP, colored by OPCST, Age, Tissue origin, and Tissue location. Each point represents a trajectory. **(G, J)** GO enrichment of Trajectory related Replicable Axis (TRAV) gene module, visualized with dotplot. The bottom dot annotation represents the differentiation stage at which the gene module executes its function. **(H,K)** TRAV activity with UMAP visualization in **(B-E) (I,L)** Trajectory dotplot illustrates the gene expression patterns across 20 randomly sampled trajectories from four OPCST.

module activities, making it interpretable and providing a detailed view of gene expression dynamics and gene module activity across population-level trajectories (S19A–S19E Fig and Note 7 in S1 Text). Our methods facilitate multi-scale exploration of genes and gene modules in differentiation processes.

**TRAVMap** reveals that osteoprogenitor cell identity, age, and tissue origin can all drive heterogeneity in osteoblast differentiation processes (Figs 5C–5F and S19A–S19E). Among these factors, the identity of the osteoprogenitor cells contributes the most to this observed heterogeneity. Therefore, we identified dozens of OPCST-associated gene modules. We focused on gene modules in LepR+ BMSC OPCST and found that these gene modules function at specific stages of the differentiation process, representing distinct biological processes. For example, TRAV122 functions at the early differentiation process, enriched genes associated with taxis; while TRAV143 functions at the middle stage of differentiation, enriched genes involved in the structure of the extracellular matrix (Fig 5G and 5H). To examine the expression pattern of the TRAV143 gene module, we randomly selected five trajectories for each OPCST and visualized them using a trajectory dotplot (Fig 5I). We observed that most of the genes within this module exhibited high expression that peaked at the middle stage of differentiation, specifically in the LepR+ BMSC OPCST trajectories, whereas they exhibited distinct pseudotemporal expression patterns in trajectories derived from other OPCST (Figs 5I and S20H). This confirms that the TRAV143 gene module functioned during the middle stage of differentiation, specifically in the LepR+ BMSC OPCST (Figs 5H and 5I and S20D).

In this way, we identified osteogenesis-conserved gene modules and age-related gene modules. The osteogenesis-conserved gene modules were active across most of the osteoblast differentiation trajectories, representing the conserved genetic programs driving the osteogenic process (Figs 5J and S20A). We found that most of these gene modules functioned at the late stage of differentiation, including TRAV208, TRAV77, and TRAV6, which were associated with biological processes like ossification, bone mineralization, and growth factor binding (Fig 5J). The age-related gene modules differentially activate between age groups. One example was TRAV140, which exhibited differential activity across LepR+ BMSC, Fibroblast, and Mes OPCSTs (S21A and S21B Fig). In postnatal and young adult LepR+ OPCST trajectories, TRAV140 showed high activity, and its associated gene module, containing genes like *Smpd3* and *Ano6*, reached peak expression at the late stage (S21D and S21F Fig). However, in adult and old groups, TRAV140 activity was lower, and the gene module exhibited inconsistent expression patterns (S21D and S21G Fig). Notably, a majority of genes (61 out of 100) within the TRAV140 module overlapped with genes identified by **TrajDiff**, further supporting its age-related nature (S21E Fig). Gene Ontology analysis suggested that this module was associated with bone morphogenesis and stem cell development, implying potential differences in osteogenesis activity across age groups (S21H Fig). Interestingly, the predicted transcription factors regulating TRAV140, *Fosl1*(+) [53], and *Lef1*(+) [54], represented promising therapeutic targets for age-related bone disorders like osteoporosis (S21I Fig).

In conclusion, **TRAVMap** effectively identified both osteoprogenitor-related and age-related gene modules. This allowed for the creation of a landscape of pseudotemporal gene modules that are active at distinct developmental stages across population-level differentiation trajectories.

## Applying TrajAtlas to an extended atlas elucidates alterations in trajectories under injury conditions

The **Differentiation Atlas** is primarily designed to understand the osteogenesis process in healthy mice, aiming to establish a reference model for normal osteoblast differentiation. However, osteoblast differentiation can be altered or disrupted by various conditions, including injury [7,55], and disease [55]. Projecting new datasets onto a reference atlas has been shown to effectively detect aberrant cell states [56]. To investigate osteogenesis within diverse microenvironments, we collected public datasets that incorporated a broader range of tissues

(e.g., digit bone, rib, periodontium), encompassed more complex cellular states (such as injury, heterotopic ossification, and disease), and included data from multiple species (human and rat) (S22A–S22F Fig and S1 Table). Then, we mapped our extended datasets to the Differentiation Atlas using scArches, a reference mapping approach (Fig 6A). The final integrated datasets comprises 319 samples and 781,397 cells, which we refer to as the extended atlas.

Following data integration and label transfer, most cells were confidently annotated with level-2 labels (Figs 6A and S23E). However, we observed high uncertainty associated with a specific mesenchymal stem cell state, primarily composed of cells from injury datasets (Figs 6B and S23A–S23F). Notably, this state, termed "injury Mes", appeared to originate from various injured tissues such as skeletal muscle, rib bone, and calvarial bone (Figs 6C and S23G). This suggests that cells from different tissues may be reprogrammed to a Mesenchyme-like state upon injury, reminiscent of blastema formation in salamanders [57]. In salamanders, cells can dedifferentiate and acquire pluripotency to regenerate tissues. Interestingly, in this study, injured Mes expressed typical Mes marker genes, such as *Prrx1* [29] *and Twist1* [29] (S23H and S23I Fig), but also exhibited high expression of genes related to angiogenesis, such as *Tagln2* [58], which distinguished them from typical Mes (Figs 6D and S23J). Among the tissues that give rise to injury Mes, we validated these results specifically on the long bone. We chose *Mfap4* as a marker gene, as it is specifically expressed in injury Mes in the long bone (Fig 6E). Immunohistochemistry results showed that *Mfap4* is highly expressed in the long bone after injury, and is localized near the injury sites (Fig 6F). In contrast, *Mfap4* expression was rarely detected in uninjured long bones (Fig 6F).

Building on previous research that identified osteogenic potential in cells from injured tissues like skeletal muscle [59] and calvarial bone [60], we applied the **Differentiation Model** to the extended atlas (Fig 6A). This analysis revealed a distinct trajectory for injury-derived Mes OPCSTs transitioning towards osteoblasts, deviating from the trajectory of typical Mes OPCSTs (Fig 6G). We employed **TrajDiff** to explore the heterogeneity in the differentiation processes between these two populations. The cell abundance analysis suggested that injury-derived Mes OPCSTs exhibited higher density in the early stages of differentiation, but this density was lower in the late stages (S24A Fig). This pattern was particularly evident in tissues like skeletal muscle, where cells in the late stages were scarce (S24A Fig). These findings suggest that injury-derived Mes might have a slower differentiation speed [47] than typical Mes.

Next, we performed a differential expression analysis using **TrajDiff**, which revealed distinct gene expression patterns. Genes associated with wound healing (e.g., *Mia3*, *Hif1a*, *Smoc2*) and chemotaxis (e.g., *Cxcl9*, *Lrp1*, *Ecscr*) were consistently upregulated throughout differentiation in injury-derived Mes (*Up-Up*) (Figs 6H and 6I and S24B). Conversely, genes involved in cell fate commitment (e.g., *Hes1*, *Bcl11b*, *Id2,*) were downregulated (Down-Down) (Figs 6H and 6I and S24B). This suggests that injury may steer these cells away from terminal differentiation states. Additionally, genes related to interleukin-1 beta production (e.g., *Cd36*, *Ccl3*, *Igf1*) showed early upregulation (*Up_0*), indicating a potential role in the initial response to injury (Figs 6H and 6I and S24B). Interestingly, genes involved in growth factor activity (e.g., *Ogn*, *Hbegf*, *Clec11a*) displayed a transient upregulation pattern, being upregulated early but downregulated later, suggesting a stage-specific role for growth factor activity in injury Mes differentiation (Figs 6H and 6I and S24B).

We then applied **TRAVMap** to our extended atlas (S25A–S25F Fig). This analysis identified three injury-related gene modules (S25G–S25K Fig). These modules exhibited high activity specifically in injury trajectories, primarily functioning at the early stage of differentiation (S25L Fig). GO enrichment analysis revealed that these modules (TRAV540, TRAV766, TRAV767) are associated with distinct processes: immune response (*Sema7a*, *Rsad2*, *Tek*),

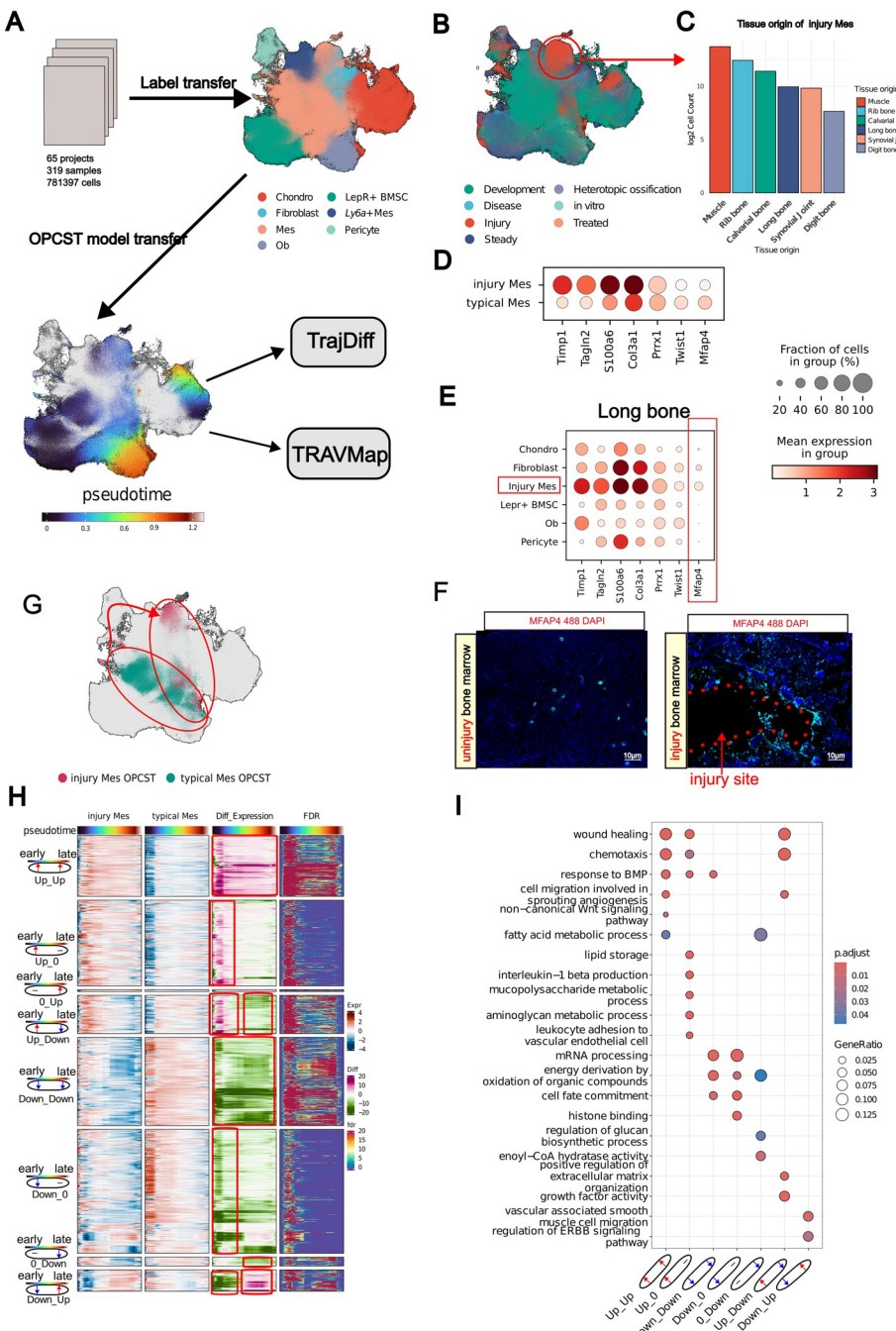

**Fig 6. Applying TrajAtlas to an extended atlas elucidates alterations in trajectories under injury conditions. (A)** An overview of extended atlas construction. **(B)** UMAP visualization of the extended atlas, colored by stage, highlighting cells (Injury Mes) that group together in the injury state. **(C)** Barplot shows the cell count (log2 scale) of tissue origin for Injury Mes in **(B)**. **(D)** Dotplot displays the expression levels of Mes marker genes and differentially expressed genes between typical Mes and injury Mes. **(E)** Dotplot shows expression levels of marker genes of injury Mes across all cell types present in long bone datasets. **(F)** MFAP4 (green) in 8-week-old femur from injured and uninjured conditions. **(G)** UMAP visualization of Mes OPCST between injury and typical states. **(H)** Heatmaps are presented in four vertical panels to illustrate gene expression of the injury Mes group (first panel) and typical Mes group (second panel), expression differences between the two groups (third panel), and FDR (fourth panel). **(I)** GO enrichment of seven clusters in **(F)**, visualized with dotplot.

bone development (*Scx*, *Acp5*, *Fam20c*), and cell proliferation (*Klf4*, *Egr1*, *Hpgd*), respectively (S25M Fig).

## Discussion

In this study, we introduce **TrajAtlas**, a novel framework centered on trajectories to analyze differentiation heterogeneity. Unlike previous methods, **TrajAtlas** prioritizes trajectories to build a comprehensive landscape of osteoblast differentiation, representing the first framework to systematically dissect large-scale trajectory heterogeneity. **TrajAtlas** allows multi-scale exploration of differentiation, examining both individual cells and entire samples. This deepens our understanding of stem cell heterogeneity and how trajectories influence cell density, gene expression, and gene modules. Our key technical innovations include: (1) integrating large-scale trajectories to construct a reference atlas and a universal differentiation model, (2) developing statistical tools to identify differentially expressed genes along trajectories, and (3) implementing methods to infer conserved pseudotemporal gene modules across population-level samples. Additionally, we established methods to visualize changes in gene expression and gene modules across population-level datasets. To guarantee the framework's universality for osteogenic processes, we initially built a core reference atlas using data from three tissues. We then successfully validated the framework on the extended atlas with various osteogenic datasets.

The major challenge in skeletal biology lies in the diversity of osteoprogenitor cells [5,6]. Due to a lack of comprehensive annotation, the characterization of responding trajectories has been a subject of controversy for a long time [5–7,25,34]. Our work addressed this gap by mapping experimentally validated osteoprogenitors onto our detailed differentiation atlas, which utilizes a seven-level annotation system. This analysis categorized osteoprogenitors into six major clusters, with at least four exhibiting the potential to directly transform into osteoblasts. We further identified key transcription factor groups that differentially regulate this transformation process, termed osteoprogenitor cell-specific differentiation trajectories. By investigating several crucial pathways for osteoblast differentiation, we revealed distinct activity patterns during osteogenesis, influenced by both the cellular source of the osteoprogenitors and the stage of differentiation. For example, the Hedgehog signaling pathway, known for its role in bone formation and maintenance [61], displayed varying activity levels during the early stages of Mes OPCSTs, with *Hhip* and *Tubb6* identified as potential target genes. This OPCST-dependent pathway activity suggested an osteoprogenitor-specific approach for selecting therapeutic targets for bone diseases [62]. Finally, by applying **TRAVMap**, we identified dozens of pseudotemporal gene modules that function sequentially throughout OPCST differentiation. This work sheds light on the dynamic interplay between genes and gene modules as differentiation progresses.

Bone formation and bone disease are closely related to age [17,34] and tissue location [5,6]. In this study, we investigated the influence of age and tissue location on bone formation and disease, focusing on four key aspects: cell type composition, osteoprogenitor state, differentially expressed genes along osteoprogenitors' differentiation trajectories, and gene modules. We identified universal age-related hallmarks across osteoprogenitors, such as cell cycle activity and Wnt pathway activation, which aligned with previous findings on senescence [36]. **TRAVMap** analysis revealed a similar conserved effect in gene modules, consistently associating specific modules with age across multiple OPCSTs. This age-related influence was likely mediated by *Fosl1*(+), suggesting potential therapeutic targets for age-related bone diseases like osteoporosis. Notably, **TrajDiff** analysis of LepR+ BMSC OPCSTs revealed differentially expressed genes categorized by their expression patterns throughout differentiation. This

finding highlights the highly specific and variable effect of age on different OPCSTs and differentiation stages.

Several tissues are widely thought to contribute to bone regeneration, including bone marrow [7], periosteum [35], and skeletal muscle [59]. By analyzing datasets from injured tissues, we identified an injury-related Mesenchyme-like cellular state. This phenomenon resembled blastema formation observed in salamanders [57] and fish [63], where blastemas arise from the dermis, cartilage, and muscle cells. Interestingly, mature osteoblasts were only observed in datasets derived from long bones and calvarial bones, suggesting a potential loss of regenerative capacity in mammals [64]. Compared to typical Mes, injury Mes exhibited altered differentiation trajectories. Using **TrajDiff**, we identified genes and gene modules associated with injury, providing valuable insights into bone regeneration processes.

While our framework is primarily designed to analyze osteoblast differentiation, it has the potential to be adapted for studying other differentiation processes, cell reprogramming, and even the cell cycle. Traditional "cell-centric analysis" offers a snapshot of cell type and gene expression at a single point in time. In contrast, "trajectory-centric analysis" sheds light on dynamic processes by examining changes over time. In summary, our study introduces a novel trajectory-centric framework that provides new insights into the dynamic interplay between cells, genes, and gene modules during osteogenesis.

While TrajAtlas offers a robust framework for modeling differentiation across large-scale datasets, there are still limitations to address. First, although our reference atlas includes datasets from a variety of tissues, it lacks finer spatial resolution. This finer resolution is crucial for understanding differentiation within specific niches and for elucidating processes such as cell communication [20] and angiogenesis [65] that are related to differentiation. However, with the advancements in spatial omics, constructing a more detailed spatial atlas [66] soon seems promising. Second, our atlas primarily focuses on the transition of different cell states into osteoblasts. However, this transition is likely not unidirectional [5]. For example, bone marrow stromal cells (BMSCs) can differentiate into chondrocytes [25], and chondrocytes can also differentiate to BMSCs [67]. The multidirectional transitions among various osteoprogenitors are also critical aspects of osteogenesis. Given that single-cell datasets alone make it challenging to recover these multidirectional cellular transitions, incorporating sequencing-based lineage-tracing methods [68] holds promise for developing more complex models that can reveal intricate differentiation dynamics soon.

## Materials and methods

### Ethics statement

The data were directly obtained from published studies, with each study appropriately referenced. Details regarding ethics approval are provided within each manuscript. All of the procedures involving mice were approved by Wuhan University (protocol NO. S07923030G).

### Dataset collection

**Data source.** All datasets were derived from published studies. In **Differentiation Atlas**, to establish a representative landscape of osteoblast differentiation, datasets satisfied the following requirements were included: 1. Samples were collected from three tissue origins: the head, limb buds, and long bones, which serve as the primary sources of osteoblasts in mice [5,6]. 2. Samples were collected from healthy mice without additional treatment. 3. Representative of a specific aspect of osteoblast formation. For example, GSE190616 [28] was selected for inclusion due to its derivation from *Col10a1*-cre mice, which represents a pivotal

transformation from hyperchondrocytes to osteoblasts. After selection, the **Differentiation Atlas** comprises datasets from 26 studies (S1A–S1C Fig and S1 Table).

Single-cell datasets related to osteogenesis were included in the extended atlas, to capture the diversity of osteogenesis. The extended atlas encompasses a broader range of tissues, including digit, rib bone, periodontium, and skeletal muscle; a wider array of treatments, such as disease models, drug interventions, and gene knockdown experiments; and multiple species, including human and rat models. Finally, 66 studies were included in the extended datasets (S22A–S22F Fig and S1 Table).

**Metadata collection.**  For each study sample, we collected metadata, including information on the age, tissue origin, sequencing method, species, genotype, treatment methods, tissue dissociation methods, associated literature, and GEO accession (S1 Table). We divided age into six groups, based on previous standards and osteogenesis changes. Organogenesis stage (E8.5-E14) [69], Fetal stage (E14.5-E18.5), Postnatal (P0-P30), Young adult (1M-3M) [34], Adult (3M-18M), Old (>12M) [70]. We divided tissue origin into head, limb bud, and long bone. According to previous studies, these three tissues represent three different sources of osteoblasts. Specifically, osteoblasts in the head region are derived from mesenchyme and fibroblasts, those in the limb bud from perichondrium and chondrocytes, and those in the long bone from chondrocytes and BMSC [5,6]. The detailed description was provided in Note 1 in S1 Text.

## Preparation of differentiation atlas construction

**Data preprocessing.**  We performed two rounds of preprocessing.

In the first round of preprocessing, our objective was to filter out cells unrelated to osteogenesis and conduct annotation for integration in every dataset separately for each dataset. In detail, we excluded cells with fewer than 300 RNA features and fewer than 800 RNA counts. Additionally, cells with a high proportion (>20%) of transcript counts derived from mitochondrial-encoded genes were removed. Genes present in fewer than 3 cells in the datasets were also eliminated. We utilized **Seurat**'s normalization method, LogNormalize, with a scale factor set to 10,000. The FindVariableFeatures function in **Seurat** was employed to select the top 2000 highly variable genes for downstream analysis. Principal Component Analysis (PCA) was applied for dimensionality reduction, considering the top 30 dimensions for analysis. For studies involving multiple samples, batch effects were corrected using the **Harmony** algorithm [71]. We employed the **Louvain** algorithm for constructing a shared nearest-neighbor graph and conducting clustering analysis, implemented in the FindNeighbors and FindClusters functions. Subsequently, we visualized the identified clusters using the UMAP method. We performed two types of annotations. The first annotation was based on the study from which the datasets originated. The second annotation was conducted based on prior knowledge, manually assigning cells to eight categories based on marker genes (Note 2 in S1 Text). After the annotation process, we excluded cell types that were not related to osteogenesis, such as epithelial cells, from further analysis. Cell clusters meeting either of the following criteria were considered relevant to osteogenesis and included in the differentiation atlas: 1. Literature reports indicating a clear ability to form osteoblasts, such as chondrocytes. 2. Continuous adjacency with osteoblast cells on the reduction of diffusion map or UMAP.

The second round of preprocessing included droplet detection, count normalization, and highly variable gene selection. Specifically, we employed **scDblFinder** [72] for droplet detection, removing cell barcodes marked as doublets from the matrix. To address differences in total UMI counts per cell, we conducted **SCRAN** normalization, following the procedure described in the Human Lung Cell Atlas (HLCA) [56]. The process involved total count

normalization, log transformation, PCA, neighborhood graph calculation, **Louvain** clustering, and subsequent **SCRAN** normalization on raw counts using **Louvain** clusters. The resulting size factors from this process were utilized for normalization. For the final datasets, cells with unusually low size factors or exceptionally high total counts after normalization were excluded from the data. Finally, we made highly variable genes selection by **Scanpy**'s highly_variable_-genes function [73].

**Batch division.** Before integration, deciding which datasets should be divided into batches is crucial to preserve biological variance and eliminate technical noise. We employed **kBET** [74], a tool based on k-nearest neighbors to quantify batch effects and determine which studies should be split into multiple batches.

**Integration method benchmarking and parameter selection.** The effectiveness of integration relies on several factors, including the integration tools, data normalization methods, the selection of highly variable genes, parameters of the integration tool, and cell annotation (especially for integration algorithms dependent on cell type information). To choose the optimal integration method for the Differentiation Atlas, we conducted benchmarking on the core datasets using the **sciB-pipeline** benchmarking framework [75].

For integration methods, we considered 12 alternatives, including **Scanorama** [76], **Harmony** [71], **bbknn** [77], **scANVI** [24], among others. Annotation categories were classified as complete annotation, no annotation, and incomplete annotation (Osteoblast and Non-osteoblast). Therefore, we performed a total of nineteen combinations, considering data normalization, integration tools, and annotation methods (S2A Fig). Finally, we utilized **scib-metric** to measure the extent of batch effect removal and preservation of biological variation for each combination. In the end, we opted for **scANVI** as the final integration solution due to its high-performance score (S2A Fig).

We employed the scvi.autotune.ModelTuner function to select hyperparameters, including n_hidden and gene_likelihood. Additionally, **scib-metrics** was utilized to determine the optimal hyperparameters for the number of variable genes and n_latent (S2B Fig).

## Construction of Differentiation Atlas

**Integration of differentiation atlas with scANVI.** For the integration of datasets into **Differentiation Atlas**, incompleted annotation (Osteoblast, Non-osteoblast) was used for integration benchmarking. **scANVI** was run on raw counts of 1500 highly variable genes (HVGs) (S2B Fig). The batch variable was described in *batch division*. While running **scANVI**, the following parameters were used: n_latent:15, encode_covariates: True, use_layer_norm: both, use_batch_norm: none, gene_likelihood: zinb, n_layers:1, dropout_rate: 0.1,max_epochs: 100 (S2B Fig).

**Cluster detection and annotation.** In general, we employed **scHarmonization** pipeline [78] (https://github.com/lsteuernagel/scHarmonization) for multi-level clustering. Initially, we applied the **Leiden** algorithm for clustering, incrementing the clustering resolution from 0.001 to 50. The Leiden clustering results with progressively increasing resolutions were selected, aiming to approximately double the number of clusters at each level. The **ROGUE** package [79] was used to evaluate the clustering results and the selection of the clustering cutoff values. The first level of clustering typically distinguishes between osteoblast and non-osteoblast cells, while the second level aims to differentiate major cell types such as chondrocytes and Mes, consistent with coarse label hierarchy. For the subsequent five levels, the **mrtree** algorithm [78] was used to form a clustering tree. The marker genes were selected using specificity scores as described in the **scHarmonization** pipeline [78]. After determining the marker genes (specificity > 1) for each cluster node in the tree, sibling clusters with fewer than 10 related

markers were merged into a single cluster node to avoid over-clustering. Visualization of the clustering tree was performed using the **ggtree** R package. The van Elteren test implementation for **Seurat** objects (https://github.com/KChen-lab/stratified-tests-for-seurat) was utilized for detecting marker genes.

We manually assigned the first three levels of annotation based on tissue origins and cell markers of cell clusters. For clusters in higher levels, we concatenated the node's best marker gene with its parent's name. The details of the seven-level annotation can be interactively explored at https://zyflab.shinyapps.io/TrajAtlas_shiny/.

**Osteoprogenitor cells incorporation.**   We collected data on experimentally validated osteoprogenitor cells (EV-OPCs) using either transplantation assays or Cre-loxP system-driven lineage tracing strategies [6] from a total of 28 studies. The collected details encompassed cell markers, publication years, study names, journals, and tissue location (S3 Table). We also collected the validation methods and categorized them into lineage tracing, sorting and transplantation, and immunostaining, corresponding to confidence levels from high to low. When annotating the osteoprogenitors in our atlas, we refer to this step as incorporation. First, we select cells from a reference atlas corresponding to a specific tissue and age. Next, we filter clusters to identify those where the marker gene is significantly highly expressed and where over 30% of the cells in the cluster express this gene. Finally, we trim the annotation tree to obtain higher-level clusters. Most marker genes of EV-OPCs (24 out of 28) are specifically expressed in our atlas, allowing them to be incorporated into our differentiation atlas. The results of OPCs incorporation can be interactively explored at https://zyflab.shinyapps.io/TrajAtlas_shiny/.

**Covariate impact on osteoprogenitor cells.**   We split covariates as technical factors (UMI per cell, number of genes detected) and biological factors (Age, Tissue) (S6A Fig). We utilized the correlation between covariates and variation in the atlas to depict how covariates influence cellular states [56]. We employed principal component regression on each covariate with scANVI latent component scores, as described in ref. [56].

Subsequently, we used the **dreamlet** package [80] to perform differential expression analysis, which applied a linear mixed model to sample-level pseudobulk. To identify age-correlated differential genes, we encoded age covariates numerically, with 'Organogenesis stage' assigned a value of 1 and 'Old' assigned a value of 6. We set "~Age" as the design formula to identify genes linearly correlated with age covariates. Genes that exhibited downregulation across all osteoprogenitors were categorized as "concordant downregulated genes", while those showing upregulation across all osteoprogenitors were termed "concordant upregulated genes". Genes that did not follow either pattern, showing mixed regulation across osteoprogenitors, were labeled as "divergently regulated genes" (S6B and S6C Fig). Gene set analysis was performed with **zenith** package (https://bioconductor.org/packages/release/bioc/html/zenith.html) with M2 gene sets in **MisgDB**.

## Lineage and pseudotime inference

**Differentiation model construction.**   The construction of the **Differentiation Model** encompassed the detection of transition probabilities, identification of endpoints, inference of paths, reconstruction of pseudotime, construction of lineage models, inference of GRNs, and analysis of genes and pathways.

We utilized **PAGA** [37] to measure coarse-grained transition probabilities between osteo-progenitors and osteoblasts. PAGA scores greater than 0.015 were defined as connected, while scores below this threshold were considered not connected.

For endpoint identification, we utilized prior knowledge, **CytoTRACE**, development time point of the samples to identify the endpoint of OPCST trajectory (Note 3 in S1 Text). Subsequently, we inferred the lineage path by **Slingshot** [23]. The pseudotime for each OPCST was individually inferred using **Palantir** [81]. To ensure the comparability of pseudotime across different OPCST, we employed the generalization capability of machine learning techniques (Note 4 in S1 Text). We utilized the **LightGBM** model due to its demonstrated excellent accuracy and generalization ability compared to other models (S10B and S10C Fig, Note 4 in S1 Text). We have shown that this model can be transferred to other single-cell datasets or bulk datasets to accurately predict the progression of osteoblast differentiation (S10F–S10H Fig and Note 4 in S1 Text).

To construct the lineage model, we partitioned each OPCST into 10 bins based on pseudotime, ensuring that each bin had an equal pseudotime gap. Subsequently, if bins at the same pseudotime across different OPCSTs exhibited a high similarity score, we merged those bins. The similarity measurement was conducted using a random forest model by assessing whether one bin could predict the other. The core idea of similarity measurement is that the more similar two cell clusters are, the harder it is to distinguish between them. In brief, the algorithm trains a RandomForest classifier on downsampled cells to predict cluster memberships. It then calculates and normalizes the predicted probabilities for each cell's classification into various clusters, generating misclassification rates that indicate the similarity between clusters based on these probabilities. In detail, we generated reference latent representations from the same cells sampled from each bin to train a random forest classifier. Utilizing misclassification rates, we computed a similarity score, reflecting the degree of resemblance between the bins.

We utilized **SCENIC** [82] to infer the transcription factor activity of each bin. We annotated each transcription factor, indicating whether it has been reported to be associated with osteogenesis (S5 Table). We utilized **tradeSeq** to build a GAM model to model pseudotemporal gene expression, then clustered genes into 12 clusters with K-means. Gene set enrichment was performed with **clusterProfiler** [83] with the Reactome database. The pathway activity was inferred by **AUCell** (https://github.com/aertslab/AUCell).

To construct a comprehensive osteogenesis gene database, we combined differential genes in our model with the Phylobone database [38], bone-related GWAS SNPs [40], bone-related genes from Gene Ontology (GO) terms [40], and bone signature genes from other publications [39,40] to annotate bone-related genes.

For the trajectory dotplot, we extracted three attributes from trajectories: correlation, peak, and expression (Note 5 in S1 Text). For "correlation", we calculated the Pearson correlation coefficient between gene expression and pseudotime. For "peak", we divided pseudotime into 10 bins and used bins that had the max expression to represent the peak. For "expression", we calculated the mean value of 10 bins as expr. We have proved that these three attributes can recover most information on expression patterns along pseudotime (Note 5 in S1 Text).

## Detecting differential abundance and expression along pseudotime (TrajDiff)

Both the differential abundance (DA) and differential expression (DE) follow this pipeline: neighborhood construction, neighborhoods differential test, and pseudotime-association test.

The benchmarking of **TrajDiff** and other differential pseudotime analysis tools (**Condiments**, **Lamian**) was described in Note 6 in S1 Text.

**Differential abundance.**   The first two steps, neighborhood construction, and local differential testing followed the methods described in **MiloR** [84]. In short, we constructed a KNN graph and defined cell neighborhoods in the neighborhood construction step. In the local

differential testing step, we counted cells in the neighborhoods to construct an $N \times S$ (neighborhood $\times$ experimental sample count) matrix, then used **edgeR** to perform differential analysis, followed by controlling the spatial FDR in neighborhoods utilizing methods described in **Cydar** [85]. In the pseudotime-association test, we projected each neighborhood on a pseudotime axis. For neighborhoods with spatial FDR below a threshold (0.05 by default), we labeled them as "Rejection"; otherwise, we labeled them as "Accept". Then we divided the pseudotime axis into $n$ intervals (100 by default) and calculated the counts of Accept ($N_{accept}$) and Rejection ($N_{rejection}$) for each interval. Then we used binomial distribution to test whether each interval had a difference as follows:

$$p_{interval} = \sum_{i=0}^{N_{accept}} \binom{N_{accept} + N_{rejection}}{i} * \lambda^i * (1 - \lambda)^{N_{accept} + N_{rejection} - i}$$

For $\lambda$ in this equation, we generated null datasets by randomly shuffling the order of samples. $\lambda$ was then calculated as $\frac{N_{accept}}{N_{accept} + N_{rejection}}$ for the null datasets. We repeated this process multiple times ($n$) to calculate the average value of $\lambda$ and improve precision. If $\frac{N_{accept}}{N_{accept} + N_{rejection}} = 0$, we set $\lambda$ to the precision of the calculation, i.e., $\frac{1}{(N_{accept} + N_{rejection}) * n}$.

For overall $p$-value($P_{overall}$), we replaced $N_{accept}$ and $N_{rejection}$ to the count of Accept and Rejection of the whole neighborhood.

We computed the mean count per million (*CPM*) within each interval derived by **edgeR** to generate $CPM_{interval}$, representing the model-fitted cell abundance. We quantified the difference in cell abundance by multiplying the log-fold change (*logFC*) and *CPM* calculated by **edgeR**, and calculated the mean value for each interval as follows:

$$DiffExpr_{interval} = mean(logFC * CPM)$$

**Differential expression.** Firstly we constructed neighborhoods as described in the section *Differential abundance*. In the neighborhoods differential test step, we made a pseudobulk for each neighborhood and constructed an $N \times S \times G$ (neighborhood $\times$ experimental sample count $\times$ gene count) matrix. Then we used **edgeR** to loop every gene to perform differential analysis, followed by controlling the spatial FDR.

In the pseudotime-association test, we employed the identical procedure outlined in the description of the *differential abundance*. We iterated through each gene to generate $P_{interval}$, $CPM_{interval}$ and $DiffExpr_{interval}$, representing the significance of differential expression, model-fitted expression, and model-fitted expression difference, respectively. The heatmap was conduct with PyComplexHeatmap [86]. GO enrichment analysis was conducted using **cluster-Profiler**. For GO-enriched genes, we used trajectory dotplots to visualize their pseudotemporal expression patterns across trajectories.

## Construction of TRAVMap

The TRAVs were identified through matrix factorization of the pseudotemporal expression matrix derived from trajectories. In detail, For each trajectory defined in **Differentiation Model** construction, we derived $CPM_{interval}$ with **TrajDiff** to form a $G \times 100$ (gene count $\times$ interval count) matrix. We filtered out empty intervals and selected the top 2000 highly variable genes (refer to *Data preprocessing*), followed by scaling matrix by row. Then we applied non-negative matrix factorization (NMF) [50] for matrix factorization, decomposing each dataset into 15 factors (the number of factors was chosen by cross-validation). Merging

all factors, we performed hierarchical clustering [51] to form 226 clusters (= round (datasets count × factor count) / 8). We calculated the mean signal of factors in each cluster denoted as TRAVs. We extracted the top 100 genes to define the TRAV gene modules. We referred to the methodology described in ref. [51] and calculated the correlation coefficient between TRAVs and the NMF factors of each trajectory, denoting it as TRAV activity. The gene expression patterns of gene modules were visualized with trajectory dotplots. We calculated the expression, peak, and correlation in the trajectories with the highest TRAV activity to represent the expression pattern of each gene module. The GO enrichment of TRAV gene modules was conducted by **clusterProfiler**. The transcription factors of TRAV gene modules were predicted using the TRRUST database via **Enrichr** (https://maayanlab.cloud/Enrichr/).

To visualize gene module activity on population-level trajectories, we utilized TRAV activities and gene expression patterns to learn trajectory representations. The detailed methods for trajectory reduction were described in the Note 7 in S1 Text.

We used the FindMarkers function in **Seurat** to detect OPCST-related TRAVs, denoting TRAVs that have high activity (0.4) in more than one-third of trajectories as osteogenesis-conserved TRAVs. We utilized a linear mixed model to detect age-related TRAVs (*Covariate Impact on Osteoprogenitors*).

## Extended Atlas

**Construction of extended Atlas.**   The preparation step closely follows the procedure outlined in the **Construction of Differentiation Atlas** section. In total, we collected 781,397 cells from 65 datasets. We then utilized the model trained on the Differentiation Atlas to learn the latent space. To transfer labels and lineages from the core differentiation atlas, we employed a KNN classifier from scArches for cell type label transfer from the reference to the query. Additionally, we transferred common pseudotime using LightGBM with the model constructed in the OPCST model.

**Novel cell state identification.**   We employed Leiden clustering methods for our analysis, adjusting the resolution between 0.01 and 15 with 24 gradients. We then use uncertainty, measured during label transfer process, to identify new cell state [56]. An uncertainty threshold was established at the 80th percentile deviation from the core datasets. Clusters exhibiting higher mean uncertainty than this threshold were identified as candidate novel clusters.

## Mice

Female C57BL/6 mice were obtained from the Animal Center of Wuhan University. All experimental procedures were approved by Wuhan University and were performed according to laboratory animal care and use guidelines. The study protocol was approved by the Ethics Committee for Animal Use of the Institute of Biomedical Sciences (Protocol number 69/2017).

For the injury model, female C57BL/6 mice (8 weeks old) were operated on for ablation surgery. Right femurs were operated, while left femurs were untreated and used as an internal control. We followed the procedure described in the previous study [7]. To create a bone marrow injury model, an incision was made on the skin, the knee ligaments were separated, and a cylindrical area of the marrow space in the femur was sequentially removed using endodontic instruments of increasing gauge sizes. The surgical site was irrigated with saline, and the incision was sutured closed.

## Histology and immunohistochemistry

For histochemical analysis, the femurs were fixed in 4% paraformaldehyde at 4°C for 24 hours, decalcified in 10% EDTA (pH 7.4) for 4 weeks, and subsequently embedded in paraffin. 5 μm

sections were prepared using a microtome (Leica). For immunohistochemical staining, the sections were digested with 0.05% trypsin at 37°C for 15 minutes. The sections were then incubated with anti-MFAP4 antibody (1:200; ThermoFisher) overnight at 4°C, followed by incubation with goat anti-rabbit Alexa Fluor 488 secondary antibody. Fluorescence was visualized using a fluorescence microscope (Thunder Imager; Leica Microsystems).

## Supporting information

**S1 Fig. Overview of differentiation atlas. A,** Heatmap visualization of cell count (second columns) and metadata of **Differentiation Atlas**. **B,C,** Barplot shows cell count ($\log_2$ scale) of (**B**) Age group and (**C**) tissue origin group. This figure was created with BioRender.com. (TIF)

**S2 Fig. Selection of single-cell integration method and scANVI hyperparameters. A**, Result of data integration benchmarking. The rows represent methods tested, using a particular preprocessing and output. Preprocessing is summarized by "Scaling" (specifying whether or not gene values were scaled to mean 0 and standard deviation 1 across cells). Methods are sorted by overall score. The overall score is a weighted mean of the batch correction score and the bioconservation score, which in turn are a mean of the individual metrics within the category. The output column specifies whether a method has corrected gene counts, an integrated embedding, or an integrated graph as output. **B,** Result of hyperparameters (HVG, n_latent) of **scANVI** benchmarking. The rows represent hyperparameters tested. **C,D,** Cell clusters annotated with previous studies are well separated with UMAP reduction in **scANVI** latent space. (TIF)

**S3 Fig. A full seven-level annotation system of the Differential Atlas. A, A full seven-level hierarchical tree of clusters of Differential Atlas.** The first five levels with up to 49 clusters are presented, emphasizing the diverse nature of OPCs across various tissues and age groups. The left heatmap (red) depicts the overlapping of experimentally validated OPCs with clusters in the lowest tree level in the **Differentiation Atlas.** The middle heatmap (orange) depicts the relative percentage contribution of each cluster at the lowest tree level to the age group. The right heatmap (dark green) illustrates the relative percentage contribution of each cluster to the tissue origin group. (TIF)

**S4 Fig. Seven-level annotation system harmonized annotations from previous studies and different tissues. A**, Sankey plot shows how level-2 annotations harmonize cell types from different tissues (LA: long bone; LE: Limb bud, C: Head). **B-H**, Barplot shows that level-2 annotations and level-3 annotations harmonize cell types from different studies. (TIF)

**S5 Fig. Examples of mapping experimentally validated OPCs to Differential Atlas. A-E,** Examples of mapping experimentally validated OPCs to **Differential Atlas**. Left panels show the information on experimentally validated OPCs. The middle panels illustrate the cells mapped based on marker expression and tissue locations with UMAP visualization. The left panels show the expression of OPCs' markers with UMAP visualization. (TIF)

**S6 Fig. Age-related variation in OPCs states. A,** Barplots show the cell proportion of Mes subpopulations across different age groups. **B**, Fraction of total inter-sample variance in the Differential Atlas embedding that correlates with specific covariates. Covariates are split into technical (left) and biological covariates (right). Cell types at second annotation levels are

shown. **C**, Heatmap shows differential expression z-statistic for genes in each cell cluster. '*' indicates study-wide FDR < 5% in all panels. Grey boxes indicates a gene did not pass the expression cutoff in that cell cluster. Genes are categorized based on whether they are concordantly different across five OPCs. **D**, Barplot shows the number of genes in three categories in (**C**). **E,** Gene set analysis using the full spectrum of test statistics shows cell type conserved signatures Study-wide FDR < 0.05 is indicated by '*'.
(TIF)

**S7 Fig. Differentiation path reconstruction in Differentiation Model. A,F,K,N,** Force-directed graph visualization of differentiation path of (**A**) Chondrocyte OPCST, (**F**) LepR + BMSC OPCST, (**K**) Fibroblast OPCST and (**N**) Mes OPCST. **B,G,** Force-directed graph highlighted lineage-tracked cells of (**B**) *Col10a1*, (**G**) *Cxcl12*, **C-E,H,J,L-M,O-P** Force-directed graph visualization of gene expression that is high along the differential path.
(TIF)

**S8 Fig. Cell transitions between OPCs and osteoblasts were captured by PAGA connectivity structures. A-C,** Transition processes from specific osteoprogenitors are similar across samples.**D**, Heatmap shows PAGA connectivity between four OPCs (column) and osteoblasts in samples (row).
(TIF)

**S9 Fig. Endpoint identification of Differentiation Model. A**, Expression of adult stem cell markers visualized with force-directed graph **B**, Barplot showing cell proportion in different ages across level-5 annotation **C,** Vlnplot shows the developmental potential predicted by **CytoTRACE**. Lower values indicate higher potential.
(TIF)

**S10 Fig. Common pseudotime predicted by the LGBMR Regressor can effectively reconstruct osteoblast differentiation progression. A**, Expression of adult stem cell markers visualized with force-directed graph **B**, Barplot showing cell proportion in different ages across level-5 annotation **C,** Vlnplot shows the developmental potential predicted by **CytoTRACE**. Lower values indicate higher potential.
(TIF)

**S11 Fig. Construction of Differentiation Model. A,**Diagram of the three hypotheses. **B-D,** Force-directed graph visualization colored by (B) 10 pseudotime bins, (**C**) pseudotime bin 4 in OPCSTs, and (**D**) pseudotime bin 5 in OPCSTs. **E-G,** Pseudotime bins across different OPCSTs with similar states are merged. **H**.,Final Differentiation Model inferred by (**E-G**). This figure was created with BioRender.com.
(TIF)

**S12 Fig. Differentiation model reveals that transcription factor activities vary across different OPCSTs. A-L**, The left panel illustrates the activity of transcription factors across four OPCSTs. The colors represent pathway activity. The transcription factors to show were selected from Fig 3E. **M,** Barplot shows that most transcription factors identified in the **Differentiation Model** were validated to be related to bone formation (S5 Table). This figure was created with BioRender.com.
(TIF)

**S13 Fig. The trajectory dotplot enables visualization of gene expression across multiple trajectories while retaining most information. A**, Sixteen attributes were extracted from gene expression along the pseudotime. **B**, The barplot shows the random forest-predicted

importance of attributes in reconstructing gene expression. **C,** Scatter plot demonstrates that selected attributes can reconstruct GAM-fitted expression (left) and raw count (right). **D-I,** Trajectory dotplot intuitively reflects the different pseudotemporal expression patterns of genes. Pseudotemporal gene expression was visualized with **CellRank** (left panel) and trajectory dotplot (right panel).
(TIF)

**S14 Fig. Differentiation model identified osteoprogenitor cell-specific pathways. A**, Multiway heatmap of changes of pseudotemporal gene expression for four OPCST. Genes to show were selected with the associationTest procedure from **tradeSeq**. Gene clusters were grouped by k-means. **B-M,** The barplots shows the top five Reactome enrichment results of gene clusters in (**a**) arranged by p-value. **N,** Pre-osteoblasts between Lepr+ BMSC OPCST and Mes OPCST exhibit different states. **O,P,** Barplots show the top five Reactome enrichment results of differential genes between two states of pre-osteoblast in (**N**) arranged by p-value **Q,** Venn plot shows the major proportion of differential genes identified in **Differentiation Models** are annotated in other bone databases (S6 Table).
(TIF)

**S15 Fig. Benchmarking results of differential cell abundance analysis. A-C,** Overview of benchmarking strategy for (A) accuracy, (B) specificity, and (C) local variation detection. **D**, Heatmap reveals that **TrajDiff** can identify subtle changes in cell abundance. **E**, Heatmap reveals that **TrajDiff** exhibits robustness to false positives. **F**, Heatmap reveals that **TrajDiff** identifies local variations in cell abundance. **G-I, TrajDiff** demonstrates both high accuracy and specificity compared to **Lamian** and **Condiment.**
(TIF)

**S16 Fig. Benchmarking results of differential expression analysis. A-C,** Overview of benchmarking strategy for (**A**) mean difference, (**B**) trend difference, and (**C**) generality. **B**, **TrajDiff** performs better in detecting mean difference and trend difference than **Lamian**. **C**, TrajDiff takes significantly less time compared to **Lamian.**
(TIF)

**S17 Fig. TrajDiff detects genes that exhibit transient differential expression. A**, Heatmaps are presented in four vertical panels to illustrate transit differential genes expression of the Young group (first panel) and the Adult group (second panel), expression differences between the two groups (third panel), and FDR (fourth panel). In each panel, rows represent genes, while columns represent pseudotime. Genes are categorized into 8 clusters based on their expression patterns. **B**, GO enrichment of six gene clusters in (**A**), visualized with dotplot. **C,** The number of genes in different differential patterns (persistent or transitional).
(TIF)

**S18 Fig. TrajDiff detects differential genes present across different differentiation stages. A,** GO enrichment of seven clusters in Fig 4D, visualized with dotplot. **B-I,** Pseudotemporal gene expression visualized with **CellRank** to validate differential genes in Fig 4E. **J,** Venn plot shows the overlap between differential genes in TrajDiff and differential genes provided in the previous study.
(TIF)

**S19 Fig. Trajectory reduction enables visualization of gene expression and TRAV activity on population-level trajectories. A**, A schematic for trajectory reduction. TRAV activity matrix, gene expression, gene peak, and gene correlation were treated as four modalities of trajectories, which were integrated by weighted nearest neighbor (WNN) to get trajectory

embeddings. **B-E,** Independent analysis of (**B**) peak, (**C**) expression, (**D**) correlation, and (**E**) TRAV modalities. **F-I,** Trajectory reduction after integrating four modalities with WNN, colored by (**F**) Tissue location, (**G**) Machine, (**H**) Project, (**I**) Treatment. **J-M,** Trajectory reduction enables visualization of (**J**) TRAV activities, (**K**) gene expression, (**L**) gene correlation, and (**M**) gene peak.
(TIF)

**S20 Fig. TRAVMap identifed OPCST-related TRAVs. a**, Heatmap shows activity of TRAVs (row) across trajectories (column) of four OPCST. **b,c,** TRAV activity visualized with trajectory embeddings, colored by (**b**) LepR+ BMSC OPCST TRAVs, (**c**) osteogenesis-conserved TRAVs. **d,f,** Heatmap shows pseudotemporal gene expression of (**d**) TRAV143 and (**f**) TRAV77 in the sample with the highest TRAV activity. **e,f,** Predicted transcription factors that regulate (**e**) TRAV143, (**g**) TRAV77. **h,** Heatmaps shows shows conserved pseudotemporal gene expression of TRAV143 in multiple samples.
(TIF)

**S21 Fig. TRAVMap identfied age-related TRAVs. A**, Heatmap shows the activity of TRAV140 across OPCSTs (row) and Ages (column). **B**, Violin plots show TRAV activity across OPCSTs and Ages (row). **C,** TRAV140 activity visualized with trajectory embeddings. **D,** Trajectory dotplot illustrates the genes expression patterns of the TRAV140 gene module (row) across LepR+ OPCST trajectories (column). **E,** Venn plot shows overlapping between genes in TRAV140 gene module and genes identified with **TrajDiff** in Fig 4D. **F,G,** Heatmap shows distinct pseudotemporal gene expression of TRAV140 gene module in trajectories in (**F**) postnatal group and (**G**) adult group. **H**, GO enrichment of TRAV140 gene module, visualized with dotplot. **i,** Predicted Transcript factors that regulate TRAV140.
(TIF)

**S22 Fig. Overview of extended atlas. A-F,** Barplots show cell count ($\log_2$ scale) of (**A**) Age group, (**B**) Tissue location group, (**C**) Species group, (**D**) State group, (**E**) Gene type group, and (**F**) Machine group. This figure was created with BioRender.com.
(TIF)

**S23 Fig. A Mes-like cellular state that related to injury was identified in extended atlas. a,** UMAP visualization of jointly embedded **Differential Atlas** (core) and the projected datasets (Extend) **b**, UMAP visualization of level-2 annotation in **Differential Atlas. c,** UMAP visualization of uncertainty score (Methods) **d**, UMAP visualization to highlight typical Mes and injury Mes. **g,** Heatmap visualization of cell count (fist columns) and metadata of injury Mes in (**d**). **f,h,** UMAP visualization of typical Mes marker. **i**, GO enrichment of differential genes between injury Mes and typical Mes, visualized with dotplot.
(TIF)

**S24 Fig. Identification of differential genes in Mes OPCST between injury and typical States using TrajDiff on extended atlas. A**, Heatmap illustrates difference in cell abundance along pseudotime (column) in Mes OPCST trajectories (row) between injury and typical state. The four rows of bottom annotation represent: Mean cell abundance of the two groups (row 1, row 2), Differential abundance (row 3), and False discovery rate (FDR) (row 4). **B,** Trajectory dotplot illustrates expression of GO-enriched genes from seven gene cluster in Fig 6F across Mes OPCST trajectories.
(TIF)

**S25 Fig. Identification of injury-related TRAVs using TRAVMap on extended atlas. a-f,h,** Trajectory embeddings visualized with UMAP, colored by (**a**) OPCST, (**b**) Age, (**c**) deriving

from the **Differentiation Atlas** or projected datasets, (**d**) Machine group, (**e**) Species (**f**) Tissue origin, and (**h**) State. **g,** Volcano plot shows injury-related TRAVs. **i-f,**TRAV activity visualized with trajectory embeddings, colored by injury-related TRAV. **l,** Trajectory dotplot illustrates expression of genes of TRAV 767 gene module across Mes OPCST. **m**GO enrichment of three TRAV gene modules, visualized with dotplot.
(TIF)

**S1 Table. Metadata of DIfferential Atlas and Extended Atlas.**
(XLSX)

**S2 Table. Description of clusters in Differential Altas.**
(CSV)

**S3 Table. Description of experimental validated osteoprogenitors.**
(XLSX)

**S4 Table. Cell number of gene cluster.**
(CSV)

**S5 Table. Transcription clusters list in Fig 3E.**
(CSV)

**S6 Table. Genes related to osteogenesis identified in the OPCST model, integrated with osteogenesis-related databases.**
(CSV)

**S1 Text. Supplementary notes.**
(DOCX)

## Acknowledgments

We thank Mengxun Li, Yue Zhang, Yaoyang Ge and Yihang Dai for engaging conversation during the research process. We also thank Jialin Wu, a talented artist, for creating the striking image for this study. The numerical calculations in this paper have been done on the super-computing system in the Supercomputing Center of Wuhan University.

## Author Contributions

**Conceptualization:** Yaoting Ji, Huan Liu, Yufeng Zhang.

**Data curation:** Litian Han, Hao Zeng, Xiaoxin Zhang.

**Formal analysis:** Litian Han.

**Resources:** Huan Liu.

**Software:** Litian Han, Hao Zeng.

**Validation:** Litian Han, Yiqian Yu, Yueqi Ni.

**Writing – original draft:** Litian Han.

**Writing – review & editing:** Yaoting Ji, Huan Liu.

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
