## [Decision Letter · Decision Letter 0]

13 Jul 2024

Dear Dr Yufeng,

Thank you very much for submitting your Research Article entitled 'Trajectory-centric Framework TrajAtlas reveals multi-scale differentiation heterogeneity among cells, genes, and gene module in osteogenesis' to PLOS Genetics.

The manuscript was fully evaluated at the editorial level and by independent peer reviewers. The reviewers appreciated the attention to an important problem, but raised some substantial concerns about the current manuscript. Based on the reviews, we will not be able to accept this version of the manuscript, but we would be willing to review a much-revised version. We cannot, of course, promise publication at that time.

If you decide to revise the manuscript for further consideration at PLOS Genetics, please aim to resubmit within the next 60 days, unless it will take extra time to address the concerns of the reviewers, in which case we would appreciate an expected resubmission date by email to plosgenetics@plos.org.

To resubmit, log into your Editorial Manager account and select the option 'Revise Submission' in the 'Submissions Needing Revision' folder.

We are sorry that we cannot be more positive about your manuscript at this stage. Please do not hesitate to contact us if you have any concerns or questions.

Yours sincerely,

Christine Wells

Academic Editor

PLOS Genetics

Giovanni Bosco

Section Editor

PLOS Genetics

Reviewer's Responses to Questions

**Comments to the Authors:**

Reviewer #1: The review has been uploaded as an attachement.

Reviewer #2: Here, I am providing my expert assessment of the study by Han et al. from the biological perspective in response to the editor’s request. This study presents a novel trajectory-centric package, TrajAtlas, composed of four components of Differentiation Atlas, Differentiation Model, TrajDiff, and TRAVMap, which collectively enable single-cell level deconvolution of highly complex cell lineages such as those for bone-forming osteoblasts. The motivation behind developing a new computation approach is multifold. The osteoblast lineage is unique in that it converges to terminally differentiated cells from diverse cells-of-origins in a manner opposite to the hematopoietic lineage that presents a classical stem cell model with a single master hematopoietic stem cells diverging to numerous terminally differentiated cell types. The diverse origin of osteoblasts and contiguous and indistinctive cell states of the osteoblast lineage necessitates a different approach to model single-cell trajectories. Addressing this issue by focusing on trajectories instead of cell states represents the study's major innovation. My specific comments are listed below for the authors’ consideration.

Major points:

1. The approach successfully models trajectories instead of cell states, providing novel insights into a highly complex converging cell lineage such as osteoblasts. However, the major weakness of the approach is the lack of consideration of spatial information. In matrix-bound cell types such as osteoblasts, cell states, and trajectories are substantially influenced by their locations within the microenvironment. The location of each cell is perhaps more important than the cell state in rendering cell fate decisions. Additionally, the bias is inevitably introduced during cell dissociation and isolation to generate scRNA-seq datasets. The authors should critically appraise the limitation of the current study, particularly regarding the lack of consideration of the spatial information, and discuss the future venue.

2. The authors state that they mapped experimentally validated osteoprogenitor cells onto their datasets. However, the authors did not clarify the levels of scientific rigor in mapping each cell type. For example, biological evidence from lineage tracing studies should not be treated as equal to that from immunostaining or in vitro culture of cells isolated using a set of cell markers. It is evident that in vivo lineage tracing studies present only reliable biological evidence regarding trajectories. All other approaches would introduce a substantial bias in introducing the outcome. The authors should emphasize the outcomes of in vivo lineage tracing studies as the most rigorous biological evidence and reframe the rationale behind the study as appropriate.

3. The abstract needs a significant rework to explain the motivation behind developing a new approach and emphasize its novelty.

Individual points:

4. Abstract:

a. It is not clear what the authors mean by saying "heterogeneity of the osteoblast differentiation process." To my understanding, it is not the heterogeneity of the process per se but the diverse cells-of-origins feeding into osteoblasts through different routes that necessitate a new computational approach.

b. "Osteoporosis and bone regeneration" is somewhat an overstatement, as they did not directly test these models. It should be changed to "osteoblast differentiation."

5. Introduction:

a. The question needs to be better laid out at the end of the first paragraph. It is not the potential overlap of osteoprogenitors that hinders our understanding of osteoblast differentiation. It is the diversity of cells-of-origins of osteoblasts that necessitates the development of a new approach. Although relatively well-performed, the authors' new approach could only address one aspect without addressing more important spatial aspects.

b. It is unclear what the authors mean when they say, "overlooks the true cellular origins of osteoblasts." There is no single cellular origin for osteoblasts, as demonstrated by lineage tracing studies.

6. Results:

a. Can the authors elaborate on this statement? "Diverse osteoprogenitor populations arranged radially around osteoblasts, suggesting a potential transition between osteoprogenitor cells and osteoblasts."

b. Figure 2c: Does this amalgamated dataset include periosteal cells? If not, this would represent a significant limitation of the current study. The authors should clarify.

c. RE: osteogenesis in the resting zone. Osteogenesis does not occur in the resting zone. It occurs adjacent to the hypertrophic zone.

d. The authors need to treat potential markers for MSCs with more caution. For example, MSCs should not comprise the vast majority of the cell population – if it does, they should no longer be called "stem cells." The authors would need a different and more accurate nomenclature. There is no consensus that Msx2 or Ly6a can be globally used as an adult stem cell marker.

e. A significant overlap between LepR+ and Cxcl12+ BMSCs has been well-recognized in the field.

f. It should be noted that Hoxa11(+) cells only exist in the zeugopod (ulna, radius, tibia, and fibula).

g. Is TRAV a particular class of a metagene? If so, it should be clearly stated. If not, the authors need to explain further how it relates to a metagene.

**Have all data underlying the figures and results presented in the manuscript been provided?**

Reviewer #1: Yes

Reviewer #2: Yes

PLOS authors have the option to publish the peer review history of their article (what does this mean?). If published, this will include your full peer review and any attached files.

Reviewer #1: **Yes: **Jiadong Mao

Reviewer #2: **Yes: **Noriaki Ono

---

## [Decision Letter · Decision Letter 1]

26 Sep 2024

Dear Dr Zhang,

Thank you very much for submitting your Research Article entitled 'Trajectory-centric framework TrajAtlas reveals multi-scale differentiation heterogeneity among cells, genes, and gene modules in osteogenesis' to PLOS Genetics.

The manuscript was fully evaluated at the editorial level and by independent peer reviewers. The reviewers appreciated the attention to an important topic but identified some concerns that we ask you address in a revised manuscript.

We therefore ask you to modify the manuscript according to the review recommendations. Your revisions should address the specific points made by each reviewer.

To resubmit, log into your Editorial Manager account and select the option 'Revise Submission' in the 'Submissions Needing Revision' folder.

Yours sincerely,

Christine Wells

Academic Editor

PLOS Genetics

Giovanni Bosco

Section Editor

PLOS Genetics

Reviewer's Responses to Questions

**Comments to the Authors:**

Reviewer #1: Review attached

Reviewer #2: The authors have meticulously addressed my comments and revised the manuscript in a satisfactory manner. I do not have any additional comments.

**Have all data underlying the figures and results presented in the manuscript been provided?**

Reviewer #1: Yes

Reviewer #2: Yes

PLOS authors have the option to publish the peer review history of their article (what does this mean?). If published, this will include your full peer review and any attached files.

Reviewer #1: **Yes: **Jiadong Mao

Reviewer #2: **Yes: **Noriaki Ono

---

## [Editor Report · Decision Letter 2]

7 Oct 2024

Dear Dr Zhang,

We are pleased to inform you that your manuscript entitled "Trajectory-centric framework TrajAtlas reveals multi-scale differentiation heterogeneity among cells, genes, and gene modules in osteogenesis" has been editorially accepted for publication in PLOS Genetics. Congratulations!

Yours sincerely,

Christine Wells

Academic Editor

PLOS Genetics

Giovanni Bosco

Section Editor

PLOS Genetics

Comments from the reviewers (if applicable):

**Data Deposition**

http://datadryad.org/submit?journalID=pgenetics&manu=PGENETICS-D-24-00584R2

**Press Queries**

---

## [Editor Report · Acceptance letter]

16 Oct 2024

PGENETICS-D-24-00584R2 

Trajectory-centric framework TrajAtlas reveals multi-scale differentiation heterogeneity among cells, genes, and gene modules in osteogenesis 

Dear Dr Zhang, 

We are pleased to inform you that your manuscript entitled "Trajectory-centric framework TrajAtlas reveals multi-scale differentiation heterogeneity among cells, genes, and gene modules in osteogenesis" has been formally accepted for publication in PLOS Genetics! Your manuscript is now with our production department and you will be notified of the publication date in due course.

With kind regards,

Dorothy Lannert

PLOS Genetics

On behalf of:
